# SMALL MOLECULE OPTIMIZATION WITH LARGE LANGUAGE MODELS

## ABSTRACT

Recent advancements in large language models (LLMs) have opened new possibilities for generative molecular drug design. In molecular optimization, LLMs are promising candidates to augment traditional modeling and rule-based approaches for refining molecular structures toward design criteria. We present a novel approach to molecular optimization using LLMs trained on a hand-crafted corpus of over 100 million molecules and their properties. We trained three new models, Chemlactica-125M, Chemlactica-1.3B, and Chemma-2B, with a demonstrated ability to generate molecules with specified properties and learn new molecular characteristics from limited samples, competitive with the state-of-the-art (SOTA) in property prediction tasks on experimental data. Our optimization method, elucidated by these capabilities, combines the models' generative power with concepts from prompt optimization, evolutionary algorithms, and rejection sampling to solve molecular optimization problems more efficiently. The approach surpasses previous SOTA results on the Practical Molecular Optimization (PMO) benchmark and exceeds or is competitive with the SOTA in multi-property optimization tasks involving docking simulations. We release the training data, language models, and optimization algorithm to facilitate further research and reproducibility.

## 1 INTRODUCTION

Molecular optimization is a cornerstone of drug discovery, involving identifying compounds with specific desirable properties(Hughes et al., 2011). This process traditionally requires extensive laboratory experimentation, making it time-consuming and costly. Computational methods have emerged as powerful tools to accelerate this process, yet they often struggle with the vast and discrete nature of chemical space (Wu et al., 2018; Schneider, 2018).

Large language models (LLMs) have recently demonstrated remarkable capabilities across various domains, from natural language processing to code generation (Brown et al., 2020; OpenAI, 2023; Zhang et al., 2023). While there have been initial attempts to apply LLMs to chemical tasks (Irwin et al., 2022; Edwards et al., 2022; Chilingaryan et al., 2024), these efforts were often limited in scope or performance. Our work represents a significant leap forward, leveraging the full power of LLMs to revolutionize molecular optimization for drug discovery.

We present a novel approach that harnesses LLMs to generate and optimize small molecules with unprecedented efficiency and accuracy. Our method uniquely combines LLMs' generative capabilities with evolutionary strategies, enabling more effective exploration of chemical space than traditional graph-based or SMILES-based models.

Our research makes several contributions to the field:

1. We develop a comprehensive molecular corpus derived from PubChem (Kim et al., 2015), encompassing over 110 million molecules and their properties. This corpus, richer in chemical information compared to SMILES-only corpora used in previous studies, serves as the foundation for training our specialized LLMs: Chemlactica (125M and 1.3B parameters) and Chemma (2B parameters). These models demonstrate a deep understanding of molecular structures and properties, enabling more accurate predictions and generations.

2. We illustrate the adaptability of our models through efficient fine-tuning for various molecular property prediction tasks. With just a few hundred training examples, our models achieve

competitive performance on standard benchmarks like ESOL and FreeSolv, showcasing their potential for rapid adaptation to new tasks in drug discovery pipelines incorporating experimental data.

3. We introduce a new molecule optimization algorithm that unifies concepts from genetic algorithms, rejection sampling, and prompt optimization. This algorithm leverages our trained LLMs to navigate the vast chemical space efficiently, generating molecules with targeted properties with high sample efficiency. It achieves state-of-the-art performance on multiple molecular optimization benchmarks. On the PMO benchmark tasks (Gao et al., 2022), we achieved an average improvement of 8% over the previous best method. In drug discovery case studies involving protein-ligand docking, our method generates viable drug candidates up to 4 times faster than existing approaches.

## 2 RELATED WORK

**Language Models for Molecular Representation**   While graph-based representations are common for molecules, string-based representations, particularly the Simplified Molecular Input Line Entry System (SMILES) (Weininger, 1988), have gained new traction in molecular modeling due to their compatibility with language models(Guo et al., 2023c; Ramos et al., 2024). This approach leverages the power of pretrained language models and enables efficient processing of molecular data. Notable examples include ChemFormer (Irwin et al., 2022), MolT5 (Edwards et al., 2022), BARTSmiles (Chilingaryan et al., 2024), and LM-Desgin for proteins (Zheng et al., 2023), which adapt traditional language model architectures to chemical tasks. These models demonstrate the potential of applying natural language processing techniques to molecular design and property prediction.

**Molecular Optimization Techniques**   Molecular optimization, a key challenge in drug discovery(Schneider and Fechner, 2005; Zhou et al., 2019), involves navigating a vast combinatorial space of potential drugs while satisfying multiple constraints. Traditional approaches include genetic algorithms adapted for molecular graphs (Yoshikawa et al., 2018) and Monte Carlo tree search over molecular graphs (Jensen, 2019). Recent methods increasingly make use of machine learning, especially deep learning techniques(van Tilborg et al., 2024). For instance, variational autoencoders (Kingma and Welling, 2013) have been applied to generate and optimize molecules in latent space, including (Gómez-Bombarelli et al., 2018) and (Jin et al., 2018). The GFlowNet (Bengio et al., 2021) represents a novel approach designed to sample compositional objects (like molecules) with reward-proportional probability, making it well-suited for optimization tasks. Extensions of GFlowNets (Kim et al., 2024) incorporating genetic search have shown promising results in molecular optimization.

**Large Language Models in Optimization**   The success of large language models (LLMs) has led to their application in various optimization tasks beyond text generation. For instance, Chen et al. (2023) combined prompt tuning with evolutionary algorithms to design neural network architectures, outperforming human experts on specific tasks. Similarly, EvoPrompt (Guo et al., 2023b) developed a general evolutionary algorithm using language models, optimizing task-specific prompts for various downstream applications. Recently, generalized methods from discrete prompt optimization have been introduced(Guo et al., 2023a). Another work where optimization is performed via human-machine dialogue with a finetuned LM is DrugAssist (Ye et al., 2023). Our method uniquely combines discrete prompt optimization style evolutionary methods and LLM-based optimization to the molecular optimization problem setting. The work most similar and parallel to ours is (Wang et al., 2024), which also combines evolutionary algorithms with prompt tuning for molecular generation but does not train models on custom corpora or perform additional finetuning during optimization as is done in this work. These studies demonstrate the potential of LLMs in complex optimization problems, paving the way for their application in molecular design and optimization.

Our work builds upon these foundations, uniquely combining the strengths of large language models with evolutionary strategies for molecular optimization. We extend the application of LLMs beyond simple property prediction or generation, developing a comprehensive framework for navigating the complex landscape of molecular design.

[WEIGHT]180.16[/WEIGHT][TPSA]63.60[/TPSA][CLOGP]1.31[/CLOGP]
[START_SMILES]CC(=O)OC1=CC=CC=C1C(=O)O[END_SMILES]
[SAS]1.58[/SAS][QED]0.92[/QED]
[SIMILAR]O=C(Oc1ccccc1C(=O)O)c1ccccc1O 0.59[/SIMILAR]
[SYNONYM]aspirin[/SYNONYM][PROPERTY]Vapor Pressure
2.52X10-5 mm Hg at 25 °C (calc)[/PROPERTY][CID]2244[/CID]

Figure 1: Aspirin's molecular structure (left) and it's representation in our dataset (right)

## 3 TRAINING CORPUS

For our training data, we extract information on molecules from PubChem(Kim et al., 2015), encompassing information on the molecules, similar molecule pairs, experimental properties, and bioassays; we store the data in a database. We randomly reserve approximately 10,000 molecules to monitor during training and for select experiments in 4, where the validation set is mentioned. We used *rdkit* (Landrum et al., 2013) to compute key molecular properties, get more precise similarity measurements, and standardize SMILES strings. We transformed our database into a corpus of molecular documents with key-value pairs representing identifiers and information for a given molecule. To provide this as input to the language models, we developed a template system using paired tags to delimit each property and data point for the final string representation of molecules that we derive from their intermediate key-value representations. For instance, the string representation for a molecule's quantitative estimated drug-likeness (QED) value is `[QED]0.84[/QED]`. To enable property prediction and property-conditioned molecular generation, we randomized the property order and either set the position of the primary molecule at the start of the document or in between other tags with equal probability. Figure 1 illustrates the document corresponding to aspirin. We provide more details on the training data in the appendix section A.4 and open-source the key-value representation of our data on huggingface.

## 4 MODEL TRAINING AND EVALUATION

**Selection of Pretrained Language Models**  We chose models for continued pretraining based on their general-purpose performance and domain-specific knowledge. At its release, Galactica (Taylor et al., 2022) outperformed models like OPT (Zhang et al., 2022), Chinchilla (Hoffmann et al., 2022), and BLOOM (Workshop et al., 2022) on tasks such as BIG-bench (bench authors, 2023), MMLU (Hendrycks et al., 2020), and TruthfulQA (Lin et al., 2021). Its pretraining included two million PubChem molecules, SMILES-specific tagging, and a scientific corpus, making it well-suited for molecular data. Gemma (Team et al., 2024), while not explicitly trained on molecular data, underwent extensive pretraining (2 trillion tokens for Gemma-2B) and demonstrated state-of-the-art performance on benchmarks like MMLU, HellaSwag (Zellers et al., 2019), and Human eval (Chen et al., 2021), comparable to larger models like LLaMA 2 (Touvron et al., 2023) and Mistral 7B (Jiang et al., 2023). We used the Galactica and Gemma tokenizers with minor modifications, and performed standard language model training for both models. Additional details on tokenization and training are supplied in appendix sections A.5.

### 4.1 BENEFITS OF CONTINUED PRETRAINING

To assess the efficacy of continued pretraining, we conducted two experiments designed to demonstrate (a) the potential advantages of initiating from a pretrained model versus training from scratch and (b) whether our continued pretraining methodology enhances molecular comprehension. For the first experiment, we randomly initialized a 125M parameter Galactica model and trained it following the protocol recommended in the Galactica paper (Taylor et al., 2022) on our entire dataset. The results of the downstream evaluations, presented in Table 1, demonstrate that the randomly initialized model yielded inferior performance on all conditional generation and property prediction tasks

Table 1: RMSE (RSME corrected for mean) ↓ for Property Prediction (PP) and Conditional Generation (CG) for different tasks and models.

| | QED | | SIM | | SAS | |
|---|---|---|---|---|---|---|
| | PP | CG | PP | CG | PP | CG |
| Random Init-125M | 0.030 | 0.110 (0.119) | 0.068 | 0.239 | 0.092 | 0.442 (0.736) |
| Chemlactica-125M | 0.016 | 0.084 (0.084) | 0.046 | 0.181 | 0.082 | 0.432 (0.432) |
| Chemlactica-1.3B | 0.004 | 0.069 (0.096) | **0.043** | 0.172 | 0.064 | 0.281 (0.281) |
| Chemma-2B-2.1B | 0.015 | 0.095 (0.095) | 0.049 | 0.147 | 0.057 | 0.482 (0.482) |
| Chemma-2B-39B | **0.003** | **0.059 (0.059)** | 0.045 | **0.167** | **0.049** | **0.277 (0.277)** |

| | CLOGP | | TPSA | | WEIGHT | |
|---|---|---|---|---|---|---|
| | PP | CG | PP | CG | PP | CG |
| Random Init-125M | 0.543 | 1.096 (1.096) | 2.415 | 9.628 (10.218) | 11.934 | 58.091 (58.091) |
| Chemlactica-125M | 0.101 | 0.429 (0.482) | 1.326 | 7.876 (8.774) | 6.996 | 17.267 (17.267) |
| Chemlactica-1.3B | 0.094 | 0.507 (0.507) | 1.032 | **5.579 (5.579)** | 5.699 | **13.494 (13.679)** |
| Chemma-2B-2.1B | 0.055 | 0.700 (0.700) | 1.662 | 6.307 (6.307) | 4.187 | 19.565 (19.565) |
| Chemma-2B-39B | **0.037** | **0.454 (0.454)** | 0.933 | 7.091 (7.091) | **0.640** | 15.429 (15.429) |

compared to Chemlactica-125M, which leveraged Galactica's pretraining. To address the second point, we performed supervised fine-tuning on a molecular property classification task using both Galactica-125M and Chemlactica-125M, demonstrating the superior capacity of our model to adapt to new downstream tasks. More details on this experiment can be found in Appendix A.7

## 4.2 Evaluation of Computed Property Prediction and Conditional Generation

To assess our models' proficiency in learning computed properties, we conducted two comprehensive experiments:

**Property Prediction** We randomly sampled a fixed set of 100 molecules from the validation set. For each property, we prompted the models with `[START_SMILES]`$m_i^{smiles}$`[END_SMILES][QED]`, where $m_i^{smiles}$ represents the SMILES string of the molecule. We then calculated the Root Mean Square Error (RMSE) between predicted and actual property values to evaluate performance.

**Conditional Generation** For each property, we sampled 100 values $v_i$ from the distribution of PubChem molecules. We then prompted the models to generate molecules with `[QED]`$v_i$`[/QED][START_SMILES]`. Using rdkit, we computed the actual property values of the generated SMILES and calculated the RMSE against the target $v_i$. To account for potential invalid generations, we compute a corrected RMSE by substituting the property values of invalid SMILES with the mean value of the respective property's distribution in our dataset.

Table 1 demonstrates the results of these experiments for the three models and a compute-controlled version of the Chemma-2B model (Chemma-2B-2.1B) and another controlled for the number of molecules to match the 125M model trained on the full training set(Chemma-2B-39B). We have utilized several techniques, including Chain-of-Thought (CoT)(Wei et al., 2022), repetition penalty(Keskar et al., 2019), and undesired token suppression to enhance the quality of generations. The details of these techniques, alongside an ablation study of the effect of each on generations, are included in Appendix A.6.3. Furthermore, we show that our models are well calibrated in predicting these properties in A.12.3. These experiments comprehensively demonstrate our models' capabilities in predicting molecular properties and generating molecules with specified properties. These are crucial tasks in molecular design and will become the building blocks for our optimization algorithm.

**Supervised Fine-Tuning** A notable capability of our models is their adaptability to new datasets and ability to learn novel molecular properties through supervised fine-tuning. To assess this feature, we fine-tuned our models on 6 ADME tasks introduced by Fang et al. (2023a) and 3 others from

Table 2: Regression tasks from MoleculeNet, all values are RMSE ↓.

|  | ESOL | FreeSolv | Lipophilicity | Avg |
|---|---|---|---|---|
| MoleculeNet GC | 0.970 | 1.400 | 0.655 | 1.008 |
| Chemformer | 0.633 | 1.230 | 0.598 | 0.820 |
| MoLFormer-XL | 0.279 | **0.231** | 0.529 | 0.346 |
| GROVER large | 0.831 | 1.544 | 0.560 | 0.978 |
| MolCLR | 1.110 | 2.200 | 0.650 | 1.320 |
| iMolCLR | 1.130 | 2.090 | 0.640 | 1.287 |
| BARTSmiles | 0.308 | 0.338 | 0.540 | 0.395 |
| Chemlactica-125M | $0.276 \pm 0.027$ | $0.312 \pm 0.016$ | $0.486 \pm 0.003$ | $0.358 \pm 0.012$ |
| Chemlactica-1.3B | $\mathbf{0.251 \pm 0.004}$ | $0.286 \pm 0.009$ | $\mathbf{0.463 \pm 0.006}$ | $\mathbf{0.333 \pm 0.005}$ |
| Chemma-2B | $0.297 \pm 0.018$ | $0.368 \pm 0.010$ | $0.531 \pm 0.014$ | $0.404 \pm 0.020$ |

MoleculeNet Wu et al. (2018). These tasks require the model to learn and predict different molecular properties, such as hydration-free energy, water solubility, and human liver microsomal stability. Our models demonstrate competitive performance, achieving state-of-the-art results for some tasks, as shown in Table 2. Unlike some comparable methods Sirumalla et al. (2024), Glenn Northcutt (2005) and Chilingaryan et al. (2024), we train and evaluate the model on regression tasks using next token prediction, without utilizing a dedicated regression head. We provide further details regarding data structuring, parameter choices, and results for ADME datasets in Appendix A.12.2.

## 5 MOLECULAR OPTIMIZATION ALGORITHM

We present a novel population-based algorithm for molecular optimization that leverages our trained language models. The algorithm addresses the challenging task of navigating the vast chemical space to find molecules with desired properties, subject to a limited evaluation budget. Formally, we define the molecular optimization problem as:

$$m^* = \arg \max_{m \in \mathcal{M}} O(m)$$

where $m$ represents a molecule, $\mathcal{M}$ is the set of valid candidate molecules (estimated to be around $10^6$ (Bohacek et al., 1996)), and $O : \mathcal{M} \to \mathbb{R}$ is a black-box oracle function that evaluates molecular properties. This oracle could represent complex processes such as lab experiments or quantum simulations.

Our approach maintains a pool of $P$ high-performing molecules and iteratively generates new candidates using a language model. It is built on three key innovations:

**LLM-enhanced genetic algorithm** We leverage our language models to generate molecules similar to the current pool. This process functions analogously to a genetic algorithm where language model generations replace traditional crossover/mutation operations. For $S$ randomly selected molecules from the pool, we generate a new molecule using the prompt:

`[SIMILAR]`$m_1^{smiles}$ `0.9[/SIMILAR]...[SIMILAR]`$m_S^{smiles}$ `0.8[/SIMILAR][START_SMILES]`

**Explicit oracle modeling** Inspired by the rejection sampling technique (Bai et al., 2022; Touvron et al., 2023), we incorporate oracle feedback directly into the language model by fine-tuning on high-performing molecules. To accomplish this, we use prompts of the form:

`[PROPERTY]`$O(m)$`[/PROPERTY][START_SMILES]`$m^{smiles}$`[END_SMILES]`

This explicit modeling allows the language model to learn the relationship between molecular structure and oracle scores, enabling more targeted generation.

Algorithm 1 presents our complete optimization procedure, which includes the initialization of an empty molecule pool, iterative generation of new molecules using the language model, evaluation of new molecules using the oracle function, updating the pool to maintain the top-P molecules, and periodic fine-tuning of the

---

**Algorithm 1** molecular_optimization

---

**Input:** $P$, $S$, $N$, $K$
Initialize an empty $Pool \leftarrow \{\}$
**repeat**
    1. Generate prompts for molecule generation.
    **for** $i = 1$ **to** $N$ **do**
        $(m_{i,1}, m_{i,2}, \ldots, m_{i,S}) \leftarrow random\_subset(Pool)$
        $p_i \leftarrow molecules2prompt((m_{i,1}, m_{i,2}, \ldots, m_{i,S}), null)$
    **end for**

    2. Generate $N$ new and unique molecules with the language model.
    $m_i \leftarrow LM(p_i), i = 1, \ldots, N$

    3. Update the pool with $m_i$s and keep only the top-$P$ molecules.
    $Pool \leftarrow Pool \cup \{m_1, \ldots, m_N\}$
    $Pool \leftarrow$ top-$P(Pool)$

    4. Fine-tune if necessary.
    **if** the best molecule (in terms of oracle score) has not improved for $K$ iterations **then**

        5. Take all the molecules from the $Pool$ with their corresponding similar molecules (using
    which they have been generated), $m_i, (m_{i,1}, m_{i,2}, \ldots, m_{i,S}), i = 1, \ldots, P$ respectively.
        $train\_samples_i \leftarrow molecules2prompt((m_{i,1}, m_{i,2}, \ldots, m_{i,S}), m_i), i = 1, \ldots, P$

        6. Train LM on $train\_samples_i, i = 1, \ldots, P$.
    **end if**
**until** optim. problem stopping condition

---

language model when progress stagnates. Algorithm 2 details our prompt construction process, which is crucial for effective molecule generation and model fine-tuning. For generation, vanilla temperature sampling is used.

We employ a dynamic fine-tuning strategy to adapt the language model throughout the optimization process. Fine-tuning is triggered if the best molecule does not improve for $K$ consecutive iterations, with the maximum number of fine-tuning rounds limited by the oracle budget. We use a learning rate scheduler with warm-up steps, and each fine-tuning step consists of multiple epochs with a portion of data reserved for validation to prevent overfitting.

Given the complexity of our algorithm, we adopt a focused hyperparameter tuning strategy, prioritizing the most sensitive parameters while keeping others fixed. This approach balances computational efficiency with optimization performance. Appendix A.6 provides the methodology and results of our hyperparameter tuning experiments. By combining these elements, our algorithm effectively leverages the power of large language models for molecular optimization, showing strong performance across a range of tasks as detailed in Section 6.

# 6 EXPERIMENTS

## 6.1 PRACTICAL MOLECULAR OPTIMIZATION

**Problem formulation.** Inspired by real-world molecular design settings Gao et al. (2022) proposes the practical molecular optimization (PMO) benchmark consisting of 23 molecular optimization problems. PMO focuses on sample efficiency, generalizability to different optimization objectives, and robustness to hyperparameter selection of molecular optimization algorithms. To assess optimization ability and sample efficiency, Gao et al. (2022) put a limit on the number of oracle calls for each task to 10000 and measures the area under the curve (AUC) of the top-10 average property values versus the number of oracle calls as the performance metric. AUC values are calculated after every 100 oracle calls, combined, and normalized to the $[0, 1]$ range.

**Our approach.** Using our proposed optimization algorithm we evaluate the Chemlactica-125M, Chemlactica-1.3B and Chemma-2B models. The optimization algorithm's hyperparameters are tuned for each model separately according to the hyperparameter tuning methodology described in (Gao et al., 2022) and A.6. For this experiment, we keep model parameters in bfloat16 for more rapid evaluation.

Table 3: PMO benchmark with Chemlactica-125M, Chemlactica-1.3B and Chemma-2B in comparison with other methods. REINVENT results are taken from Gao et al. (2022), Augmented memory is taken from Guo and Schwaller (2023a), and Genetic-guided (GG) GFlowNets are taken from Kim et al. (2024). Values are the average of 5 runs with different seeds, metric is Top-10 AUC $\uparrow \pm$ standard deviation

| | jnk3 | median1 | scaffold_hop | sitagliptin_mpo | sum of 4 | sum of 23 |
|---|---|---|---|---|---|---|
| REINVENT | $0.783 \pm 0.023$ | $0.356 \pm 0.009$ | $0.560 \pm 0.019$ | $0.021 \pm 0.003$ | 1.720 | 14.196 |
| Augmented memory | $0.739 \pm 0.110$ | $0.326 \pm 0.013$ | $0.567 \pm 0.008$ | $0.284 \pm 0.050$ | 1.916 | 15.002 |
| GG GFlowNets | $0.764 \pm 0.069$ | $0.379 \pm 0.010$ | $0.615 \pm 0.100$ | $0.634 \pm 0.039$ | 2.392 | 16.213 |
| Chemlactica-125M | $0.881 \pm 0.058$ | $0.359 \pm 0.060$ | $0.626 \pm 0.016$ | $\mathbf{0.649 \pm 0.051}$ | $2.515 \pm 0.119$ | $17.170 \pm 0.424$ |
| Chemlactica-1.3B | $0.866 \pm 0.021$ | $\mathbf{0.382 \pm 0.047}$ | $\mathbf{0.673 \pm 0.080}$ | $0.586 \pm 0.062$ | $2.506 \pm 0.155$ | $17.284 \pm 0.284$ |
| Chemma-2B | $\mathbf{0.891 \pm 0.032}$ | $\mathbf{0.382 \pm 0.022}$ | $0.669 \pm 0.110$ | $0.613 \pm 0.018$ | $\mathbf{2.555 \pm 0.099}$ | $\mathbf{17.534 \pm 0.214}$ |

**Results.** Our method performs strongly, surpassing the existing approaches. The algorithm powered by our smallest model (Chemlactica-125M) already improves over the state-of-the-art by a significant margin, with an AUC Top-10 of 17.170 (Chemlactica-125M) vs 16.213 (Genetic-guided GFlowNets). Additionally, strengthening the generator model improves the performance. Chemlactica-1.3B and Chemma-2B achieve AUC Top-10 of 17.284 and 17.534, respectively. For a more comprehensive understanding of optimization dynamics, Figures 5-7 illustrate visualizations of the optimization processes for sitagliptin_mpo task with different seeds and different models.

Furthermore, to investigate the novelty of generated molecules, we visualize the distance of the molecules generated by the model from the closest molecule in the training dataset (PubChem) throughout the optimization procedure. Figure 2 represents the molecules generated during the optimization along with their oracle scores and distances from the closest molecule in PubChem computed with Tanimoto similarity. Even though we did not explicitly guide the model to generate molecules distant from PubChem, we observe that the model generates molecules far from PubChem to optimize the given objective. In both multi-property optimization (MPO) tasks across all seeds, the model finds high-scoring molecules with less than 0.3 similarity to their nearest neighbor in PubChem, and we observe a similar pattern in nearly all MPO problems from PMO. We conclude that our method does not retrieve molecules from the training dataset and is able to explore the chemical space beyond PubChem to solve molecular design tasks.

Unlike most other methods, our language models can leverage additional information about the oracle if the oracle internally calculates common molecular properties. These properties can be explicitly written in the prompts used in the optimization loop. In Appendix A.11.2, we show that such enriched prompts can significantly improve the metrics for several PMO tasks.

## 6.2 Multi-property Optimization with Docking

### 6.2.1 Introduction to MPO with Docking

Molecular optimization tasks incorporating docking simulation evaluate a model's capability to generate viable molecules for practical drug discovery. These benchmarks assess the model's ability to generate plausible molecules that optimize docking scores (minimize docking energy) against specified protein targets while adhering to other desired structural or physicochemical constraints. The additional constraints help to prevent exploitation of the docking algorithms used. The primary objective of these benchmarks is maximizing the reward function with minimal oracle calls, emphasizing sample efficiency. Below, we present our approach and results on two benchmarks involving suites of MPO tasks with docking components, using different hyperparameter tuning approaches and evaluation metrics. We illustrate how generated molecules' docking scores change throughout the optimization process in A.13.2. In other experiments, we found that numerical precision is important for molecular optimization tasks (see A.11.3), so we keep model parameters in full floating-point precision for docking MPO experiments.

### 6.2.2 Docking MPO on DRD2, AChE, and MK2

**Problem formulation.** This benchmark was initially proposed in the Augmented Memory paper (Guo and Schwaller, 2023a). It focuses on three targets with extensive real-world applications: the dopamine type 2 receptor (DRD2), MK2-kinase, and acetylcholinesterase (AChE). To ensure the generation of realistic molecules, the oracle reward function incorporates additional constraints, including the maximization of QED and a molecular weight limit of 500 Da.

Consistent with other works, we quantify sample efficiency using two metrics: oracle burden and generative yield. Oracle burden measures the number of oracle calls required to generate N unique molecules above a

Table 4: Docking MPO experiments run with a maximum oracle budget of 5000 oracle calls. Note that both oracle burden and generative yield values are reward-threshold dependent, and mean values from the reported baseline works are reported. The numbers next to the metrics correspond to the thresholds, and the values in parentheses for oracle burden indicate how many unique molecules need to be generated. The best performance on each task-metric combination is bolded. We use the best-performing hyperparameters from the PMO benchmark.

| Metric | Target | Reinvent Baseline | Beam Structure 15 | Chemlactica 125M | Chemlactica 1.3B | Chemma 2B |
|---|---|---|---|---|---|---|
| Generative Yield 0.7 ↑ | DRD2 | $1879 \pm 16$ | $3474 \pm 158$ | $3733 \pm 512$ | $3659 \pm 288$ | $\mathbf{3848} \pm 98$ |
| | MK2 | $879 \pm 10$ | $3127 \pm 138$ | $\mathbf{3772} \pm 578$ | $3660 \pm 535$ | $3578 \pm 452$ |
| | AChE | $2437 \pm 53$ | $3824 \pm 162$ | $4108 \pm 67$ | $\mathbf{4193} \pm 128$ | $4092 \pm 284$ |
| Generative Yield 0.8 ↑ | DRD2 | $102 \pm 6$ | $1780 \pm 439$ | $2827 \pm 510$ | $2621 \pm 614$ | $\mathbf{2985} \pm 194$ |
| | MK2 | $2 \pm 0$ | $987 \pm 211$ | $\mathbf{2569} \pm 1156$ | $2216 \pm 522$ | $1058 \pm 465$ |
| | AChE | $147 \pm 11$ | $2059 \pm 327$ | $3246 \pm 168$ | $\mathbf{3652} \pm 349$ | $3096 \pm 372$ |
| Oracle burden 0.8 (1) ↓ | DRD2 | $168 \pm 149$ | $126 \pm 90$ | $20 \pm 29$ | $\mathbf{11} \pm 10$ | $74 \pm 62$ |
| | MK2 | $1724 \pm 802$ | $736 \pm 166$ | $345 \pm 312$ | $\mathbf{78} \pm 125$ | $189 \pm 278$ |
| | AChE | $83 \pm 29$ | $105 \pm 29$ | $22 \pm 28$ | $\mathbf{15} \pm 23$ | $74 \pm 72$ |
| Oracle burden 0.8 (10) ↓ | DRD2 | $883 \pm 105$ | $582 \pm 83$ | $\mathbf{114} \pm 08$ | $160 \pm 130$ | $240 \pm 11$ |
| | MK2 | Failed | $1122 \pm 154$ | $493 \pm 418$ | $\mathbf{248} \pm 261$ | $440 \pm 548$ |
| | AChE | $481 \pm 108$ | $462$ | $224 \pm 17$ | $\mathbf{91} \pm 103$ | $168 \pm 94$ |
| Oracle burden 0.8 (100) ↓ | DRD2 | $4595 \pm 0$ | $1120 \pm 25$ | $\mathbf{364} \pm 119$ | $430 \pm 250$ | $518 \pm 41$ |
| | MK2 | Failed | $2189 \pm 181$ | $865 \pm 533$ | $\mathbf{486} \pm 346$ | $934 \pm 918$ |
| | AChE | $3931 \pm 286$ | $1110 \pm 265$ | $497 \pm 58$ | $\mathbf{333} \pm 131$ | $433 \pm 143$ |

predefined reward threshold, and generative yield represents the number of unique molecules generated above a reward threshold for a fixed number of oracle calls. To maintain consistency in implementations, we adopt the molecular preprocessing, conformational generation, docking parameters, and aggregate reward function from the (Guo and Schwaller, 2023b), specifically comparing our results with the beam structure 15 method, which demonstrated superior average-case performance relative to other benchmarked methods. We used the same hyperparameters as those selected for the PMO experiment with no modifications.

**Results.** Table 4 presents our approach's performance on this benchmark. None of our models consistently outperforms the others for generative yield across the evaluated receptors. Conversely, Chemlactica-1.3B generally demonstrates superior performance for oracle burden, aside from oracle burden 1 and 10 for DRD2, where Chemlactica-125M is superior. Appendix A.13.3 shows the set of molecules generated at the beginning and at the end of the optimization trajectory for DRD2 docking. Furthermore, in A.13.2, we show that the average docking score of molecules consistently decreases throughout optimization, suggesting that the model learns to not only generate molecules that improve the scores of the easier properties but can generate molecules with low docking energies for the DRD2 pocket as well.

### 6.2.3 DOCKING MPO; HITS ON PARP1, FA7, 5HT1B, BRAF, AND JAK2

**Problem formulation.** Following Lee et al. (2024), we formulate the MPO objective as the product of a normalized docking score, normalized SA score, and QED score. The docking is performed on the `parp1`, `fa7`, `5ht1b`, `braf`, and `jak2` target proteins.

**Our approach.** To ensure comparability of the results with other methods, we employ the oracle function implementation used in Guo and Schwaller (2024), run our method with ten different seeds (0-9), and use an oracle budget of 3000 for each task. For hyperparameter tuning, we use an illustrative experiment that does not use the same oracle function as the docking MPO tasks (for a more detailed discussion on hyperparameter selection, refer to the Appendix A.6. Following Lee et al. (2024) and Guo and Schwaller (2024), we compute the Hit Ratio (%), that is, the percentage of molecules with QED > 0.5, SA score < 5, and docking score better than the median of the known actives, as well as the Strict Hit Ratio (%), which requires QED > 0.7 and SA score < 3.).

**Results.** Table 5 and 6 illustrate the results of our method in comparison with the baselines. We observe that our algorithm powered by our language models performs comparably with others. In terms of the Hit Ratio (%), our approach performs significantly better with the targets `fa7` and `jak2`, while in terms of the Strict Hit Ratio (%), it is significantly better with the target `fa7`. Notably, for the target `5ht1b`, our approach is inferior to Saturn for both metrics and shows statistically similar performance with the rest of the target proteins. The

outcomes illustrate the efficiency of our approach and its transferability to physics-based simulations using the illustrative experiment. The results also demonstrate the applicability of our method to MPO problems with a costly oracle, which are more prevalent in industrial drug discovery settings. We present analyses of generated molecule diversity in A.9.

Table 5: Comparision of our approach with other methods. The values represent the Hit Ratio (%) $\uparrow$ $\pm$ standard deviation across 10 independent runs. The results of Augmented Memory, GEAM, and Saturn are taken from Guo and Schwaller (2023a), Lee et al. (2024), and Guo and Schwaller (2024), respectively. Results within one standard deviation from the best one are bolded.

| Method | Target Protein | | | | |
|---|---|---|---|---|---|
| | parp1 | fa7 | 5ht1b | braf | jak2 |
| Augmented Memory | 16.966 ± 3.224 | 2.637 ± 0.860 | 52.016 ± 2.302 | 8.307 ± 1.714 | 21.548 ± 4.938 |
| GEAM | **45.158 ± 2.408** | 20.552 ± 2.357 | 47.664 ± 1.198 | 30.444 ± 1.610 | 46.129 ± 2.073 |
| Saturn | **57.981 ± 18.537** | 14.527 ± 9.961 | **68.185 ± 3.400** | **38.999 ± 10.114** | 60.827 ± 11.502 |
| Chemlactica-125M | 37.117 ± 7.325 | 70.392 ± 17.584 | 31.927 ± 2.697 | 30.107 ± 8.071 | 54.657 ± 12.759 |
| Chemlactica-1.3B | **52.333 ± 12.669** | **87.03 ± 5.37** | 40.541 ± 3.373 | **49.471 ± 12.055** | **75.02 ± 8.04** |
| Chemma-2B | **52.063 ± 6.935** | 79.97 ± 11.154 | 38.233 ± 5.846 | **44.81 ± 12.14** | **71.03 ± 9.074** |

Table 6: Comparison of our approach with other methods. The values represent the Strict Hit Ratio (%) $\uparrow$ $\pm$ standard deviation across 10 independent runs. The results of GEAM and Saturn are taken from Lee et al. (2024) and Guo and Schwaller (2024), respectively. Results within one standard deviation from the best one are bolded.

| Method | Target Protein | | | | |
|---|---|---|---|---|---|
| | parp1 | fa7 | 5ht1b | braf | jak2 |
| GEAM | 6.510 ± 1.087 | 2.106 ± 0.958 | 8.719 ± 0.903 | 3.685 ± 0.524 | 7.944 ± 1.157 |
| Saturn | **55.102 ± 18.027** | 13.887 ± 9.723 | **64.730 ± 3.717** | **37.250 ± 9.615** | **55.903 ± 13.613** |
| Chemlactica-125M | 28.907 ± 7.185 | 58.596 ± 18.698 | 25.523 ± 2.865 | 22.86 ± 7.239 | 45.943 ± 12.408 |
| Chemlactica-1.3B | 35.33 ± 12.837 | **68.22 ± 6.438** | 28.074 ± 3.641 | **34.583 ± 9.856** | **58.133 ± 8.404** |
| Chemma-2B | **37.117 ± 5.712** | **63.933 ± 11.794** | 26.757 ± 5.345 | **31.783 ± 10.171** | **54.79 ± 8.946** |

### 6.2.4 DISCUSSION OF MPO WITH DOCKING

Our findings validate the effectiveness of our approach, demonstrating that our models can leverage pretraining information and iterative fine-tuning to optimize complex reward functions, even with limited data not seen during pretraining. Furthermore, successfully transferring training parameters and sampling strategies from the PMO benchmark and illustrative experiment to the MPO docking tasks in 6.2.2 and 6.2.3, respectively, underscores our method's flexibility and robustness. This adaptability suggests that our approach could be particularly valuable when extensive hyperparameter tuning is impractical or undesirable. Specifically, the results demonstrate the applicability of our method to MPO problems with costly oracles, which are more prevalent in industrial drug discovery settings. However, our method does not directly generate 3D conformations used by the docking scoring function, has not universally outperformed the baselines across tasks, and is sensitive to numerical precision. Future work will further improve sample efficiency and apply the algorithm to even more challenging MPO problems inspired by applications within the industry.

## 7 CONCLUSION

This paper presents three language models: Chemlactica-125M, Chemlactica-1.3B, and Chemma-2B, trained on a novel corpus encompassing over 100 million molecules and their properties. We demonstrate the efficacy of these models on multiple tasks inspired by industrial drug design, with a particular focus on molecular optimization. Our proposed optimization algorithm combines the capabilities of language models with concepts from genetic algorithms. This approach has shown strong performance across various benchmarks, indicating its potential for addressing complex molecular design challenges. We publicly release our training corpus, pretrained models, optimization algorithm, and associated training recipes to support reproducibility, making an early step toward applying language models to chemical research. We hope our contributions provide a valuable foundation for future work in this domain, enabling new approaches for molecular design and analysis.

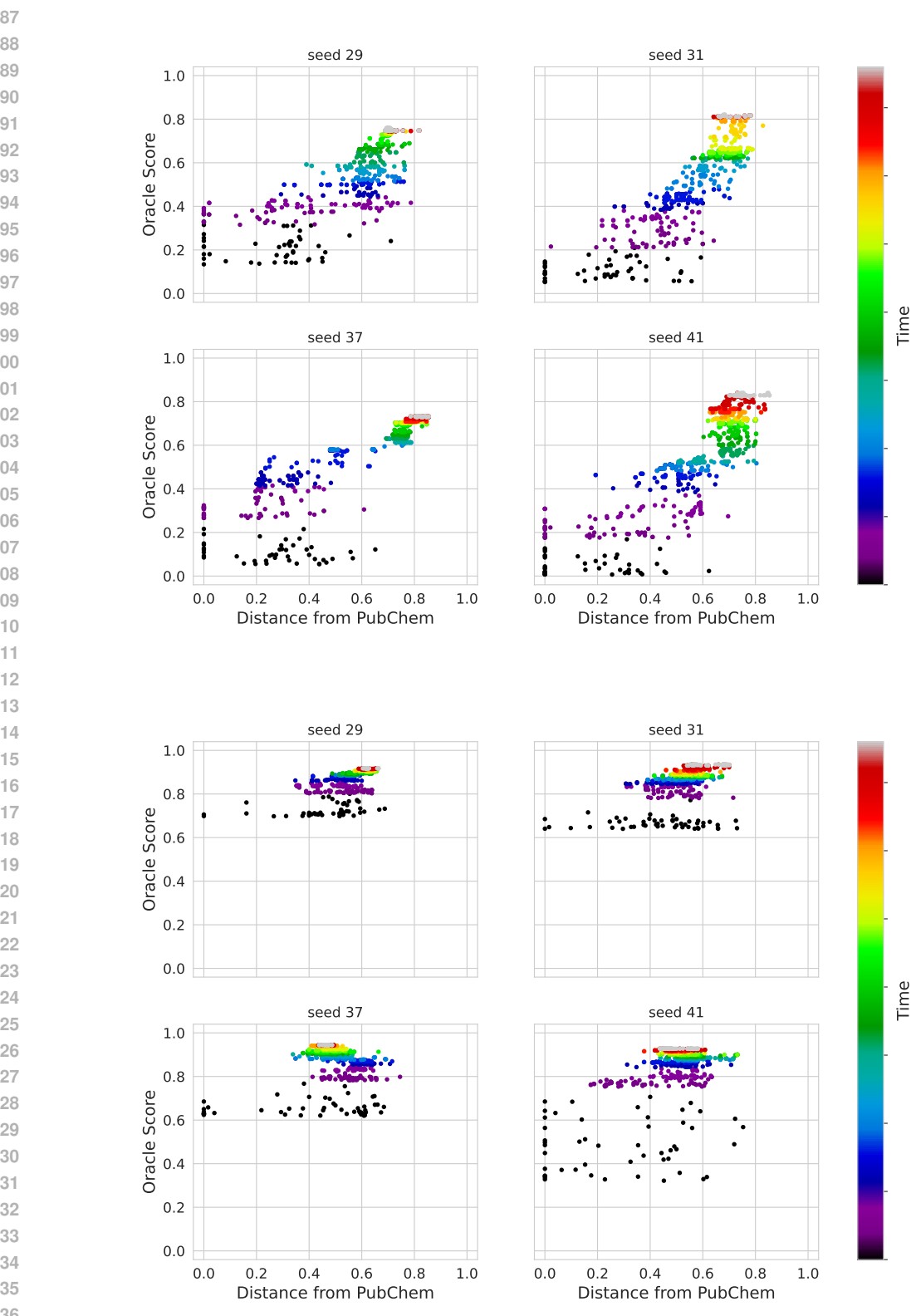

Figure 2: Visualization of the oracle score vs. distance from the closest molecule in PubChem of the current 50 best molecules throughout the optimization process. The plots are obtained for sitagliptin_mpo (top) and ranolazine_mpo (bottom) tasks from PMO benchmark with the Chemma-2B model with four different seeds.

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

# A APPENDIX

## A.1 BROADER RELATED WORK

**Recurrent Neural Networks in Molecular Design**    Recurrent neural networks (RNNs) have been applied to molecular optimization as parameter-efficient alternatives to transformer architectures. A notable example is REINVENT (Olivecrona et al., 2017), which uses policy-based reinforcement learning to generate molecules with desired properties. Another attempt at employing RNNs in drug design is ReLeaSE (Popova et al., 2018), which combines multi-phase training with small RNNs for molecular drug design. Another work that proposes a framework for drug optimization and utilizes finetuned RNNs is DrugImprover (Liu et al., 2023). Finally, recent enhancements to REINVENT, such as augmented memory and beam enumeration (Guo and Schwaller, 2023b), have further improved its performance. These approaches combine surrogate models, molecular diversity filters, experience replay mechanisms, and substructure filtering to increase sample efficiency; but these methods do not leverage the scale, generative capabilities and flexibility afforded by LLMs.

## A.2 LIMITATIONS

The language models introduced in this paper operate only on SMILES representations and do not support 3D coordinates of atoms, limiting their reliability in scenarios where 3D conformation is critical. Furthermore, the models have a very limited understanding of biological entities like proteins, constraining their practical applicability in particular biochemistry and drug discovery use-cases. While effective, the optimization algorithms presented in this paper are not exhaustively tuned, suggesting potential room for improvement. Finally, our current approach does not directly account for or consider synthetic accessibility or other practical considerations in drug design, which limits its immediate applicability in real-world drug discovery pipelines.

## A.3 BROADER IMPACT

The methods presented in this work have the potential for both positive and negative societal impacts. On the positive side, these models could significantly benefit the drug discovery and healthcare industries by accelerating the development of new therapeutic compounds. This acceleration may lead to faster responses to emerging health challenges and potentially reduce the cost of drug development.

However, as with many dual-use technologies, there is a risk that sufficiently advanced versions of these models could lower the barriers for malicious actors attempting to develop chemical or biological weapons. This risk underscores the importance of responsible development and deployment of such technologies.

Given these potential impacts, we recommend that future work in this area include rigorous evaluation of these algorithms and language models in designing potentially harmful substances to better understand and mitigate risks. Developing safeguards and ethical guidelines for using and disseminating molecular optimization models is crucial. Collaboration with experts in biosecurity and ethics will be essential to ensure that the development of these technologies proceeds in a manner that maximizes benefits while minimizing the potential for harm.

## A.4 DATASET GENERATION

**Data Collection**    We first constructed a comprehensive SQL database using PubChem dumps to generate our training corpus. Then, using *rdkit* (Landrum et al., 2013), we computed key molecular properties, including synthesizability score (SAS), quantitatively estimated drug-likeness (QED), molecular weight (MW), total polar surface area (TPSA), partition coefficient (CLogP), and various structural features such as hydrogen donors/acceptors and ring counts. Due to differences in SMILES canonicalization between PubChem and rdkit, we standardized all SMILES strings using rdkit's implementation.

Our dataset's cutoff date is January 26th, 2023, thus excluding any subsequent additions or modifications to PubChem. To ensure data integrity, molecules that failed rdkit's MolFromSmiles parsing were discarded. To incorporate similarity information, we utilized PubChem's related molecule data, which includes pairs with Tanimoto similarity $\geq$0.8 based on PubChem fingerprints. From the resulting 200 billion pairs, we sampled 4 billion and recalculated their similarities using the ECFC4 fingerprint for improved accuracy and consistency with other methods.

**JSONL Corpus Generation**    We transformed our database into a corpus of JSONL files, with each molecule represented as a single JSON object. This representation includes molecular identifiers, computed properties, similarity data, synonyms, experimental properties, and the PubChem compound identifier (CID). This representation allows for more flexible manipulation of molecules' text representation, which we describe in 3.

## A.5 TRAINING DETAILS

### A.5.1 TOKENIZATION

We utilized the original tokenizers from Gemma and Galactica, adding chemistry-specific tokens `[START_SMILES]` and `[END_SMILES]` to Gemma's tokenizer for consistency. To optimize training efficiency, we included all opening and closing tags as special tokens (e.g., `[QED]`). Samples of varying lengths were tokenized and grouped into blocks of 2048 tokens, separated by model-specific separator tokens (EOS "" for Chemlactica, BOS "<bos>" for Chemma).

### A.5.2 TRAINING IMPLEMENTATION

Chemma and Chemlactica were trained using the AdamW optimizer (Loshchilov and Hutter, 2019) with cross-entropy loss and a causal language modeling objective. We applied dropout only to Chemlactica, maintaining consistency with the original model training regimes. For computational efficiency, we train Chemma-2B in full bfloat16. We leveraged PyTorch's (Paszke et al., 2019) Fully Sharded Data Parallel (FSDP) (Zhao et al., 2023) and Flash Attention (Dao, 2024) for optimized training. The training was conducted locally (Chemlactica-125M: 306 A100 hours) and on Nebius.ai cloud (Chemma-2B: 488 H100 GPU hours, Chemlactica-1.3B: 288 H100 GPU hours). Preparatory work before the final training runs consumed multiple thousands of A100 hours.

### A.5.3 MODEL TRAINING HYPERPARAMETERS

Table 7 lists the hyperparameters we used for pretraining the aforementioned models.

## A.6 HYPERPARAMETER TUNING AND SELECTION

### A.6.1 OPTIMIZATION ALGORITHM HYPERPARAMETER TUNING FOR PMO.

Given our optimization algorithm's large number of hyperparameters, we adopt a two-phase approach. First, we identify and freeze the hyperparameters that empirically show less sensitivity to the algorithm's performance. Then, we focus on tuning the more sensitive hyperparameters using grid search. We tune the hyperparameters separately for Chemlactica-125M, Chemlactica-1.3B, and Chemma-2B to account for model-specific optimal settings. For searching the nearest neighbor with Tanimoto similarity in PubChem, we utilized the USearch similarity search engine for vectors (Vardanian, 2023).

*Selection criteria.* For tuning, we utilize the `perindopril_mpo` and `zaleplon_mpo` tasks from the PMO benchmark, following the methodology in (Gao et al., 2022). We report the AUC Top-10 metric from three independent runs with different seeds for each hyperparameter configuration.

*Fixed hyperparameters and grid.* A key hyperparameter, $N$, which determines the number of molecules generated before updating the pool, is set to 200. We implement a dynamic temperature scheduling strategy to increase the diversity of generated molecules. The sampling temperature starts at 1 and linearly increases to 1.5 as the number of oracle evaluations grows. This gradual temperature increase promotes the generation of more diverse molecules over time, reducing repetition and encouraging exploration of the chemical space.

Table 7: Hyperparameters of our language models. All cross-entropy losses use mean reduction.

|  | Chemlactica-125M | Chemlactica-1.3B | Chemma-2B |
|---|---|---|---|
| Peak learning rate | 1.4e-3 | 1.0e-4 | 1.0e-3 |
| Warmup steps | 500 | 500 | 500 |
| Context length | 2048 | 2048 | 2048 |
| ADAM $\beta_1$ | 0.9 | 0.9 | 0.9 |
| ADAM $\beta_2$ | 0.95 | 0.95 | 0.95 |
| ADAM $\epsilon$ | 1e-8 | 1e-8 | 1e-8 |
| Weight Decay | 0.1 | 0.1 | 0.1 |
| Dropout | 0.1 | 0.1 | None |
| Attention Dropout | 0.1 | 0.1 | None |
| Precision | Mixed | Mixed | BF16 |
| Loss Function | CE Loss | CE Loss | CE Loss |
| Vocabulary Size | 50066 | 50066 | 256000 |
| Gradient Clipping | 1.0 | 1.0 | 1.0 |

We perform grid search on $P$ (pool size), $S$ (number of similar molecules), $K$ (fine-tuning tolerance level), and $lr$ (fine-tuning peak learning rate) with the following grid:

- $P = [10, 30, 50]$
- $S = [0, 1, 2, 5]$
- $K = [3, 5, 7]$
- $lr = [10^{-4}, 10^{-5}]$

### A.6.2 OPTIMIZATION ALGORITHM HYPERPARAMETER TUNING FOR MPO WITH DOCKING ON PARP1, FA7, 5HY1B, BRAF, AND JAK2 TARGETS.

Consistent with the hyperparameter tuning used for PMO, we select the most sensitive hyper-parameters and then tune them separately for each model via grid search.

*Selection criteria.* Motivated by the illustrative experiment for hyperparameter tuning used in Guo and Schwaller (2023b), we formulate a problem of maximizing the TPSA of a molecule while keeping its weight under 350 Da and having more than one ring. We use this molecular design task and the oracle burden metric for hyperparameter selection.

*Hyperparameter grid.* Since the illustrative experiment takes less time to evaluate (compared to the PMO tasks), we expand the number of hyperparameters used in the grid to allow for a more extensive search.

We perform a grid search on $P$ (pool size), $S$ (number of similar molecules), $N$ (the number of molecules generated to update the pool), $temp\_schedule$ (the starting and ending temperature for linearly changing it during the optimization process), $K$ (fine-tuning tolerance level) and lr (fine-tuning peak learning rate) with the following grid:

- $P = [10, 30, 50]$
- $S = [1, 2]$
- $N = [100, 200]$
- $temp\_schedule = [[1.5, 1.0], [1.3, 1.0], [1.0, 1.0], [1.0, 1.3], [1.0, 1.5]]$
- $K = [2, 3, 5]$
- $lr = [10^{-4}, 10^{-5}]$

### A.6.3 CONDITIONAL GENERATION HYPERPARAMETERS

Our generation process benefits from the following techniques to improve output quality:

- **Chain-of-Thought (CoT):** We omit `[START_SMILES]` from the initial prompt, enabling the model to generate more property values before the molecule itself.
- **Repetition Penalty:** Applied to discourage repetitive outputs.
- **Undesired Token Suppression:** Employed to ensure the model eventually generates `[START_SMILES]`.

Table 9 provides an ablation study of these sampling components across our three models, demonstrating their individual and combined impacts on generation quality. Surprisingly, the best combination of hyperparameters, as chosen by lowest corrected RMSE (RMSE(c)), coincides with all three models. DNF (Did Not Finish) trial exceeded 30 minutes of runtime when it was manually terminated.

### A.7 CHEMLACTICA VS. GALACTICA COMPARISON

To evaluate the efficacy of our pretraining approach in comparison to the base Galactica model, we utilized the BBBP task, introduced by Wu et al. (2018). The BBBP task is a binary classification problem that requires the model to predict whether a given molecule can penetrate the blood-brain barrier. Notably, the pretraining dataset for the base Galactica model included the training set of the BBBP dataset in a question answering format, hence enabling BBBP prediction without further fine-tuning. We show the model's performance as reported in the original publication as well as our reproduction. We conducted supervised fine-tuning on base Galactica-125M and Chemlactica-125M, each using their respective data formats. It is important to note that supervised fine-tuning is necessary, as the general capabilities of our models do not inherently enable the prediction of specific downstream tasks. The results, presented in Table 8, demonstrate that although hyperparameter tuning of Galactica improved model performance significantly, Chemlactica demonstrated better performance without any tuning. With the best hyperparameters, Chemlactica's results further improve. These results suggest that our continued pretraining improves the models' ability to adapt to downstream tasks with fine-tuning.

Table 8: Comparison of Galactica-125M vs Chemlactica-125M on BBBP binary classification.

|  | ROC ↑ |
|---|---|
| Galactica-125M (original paper) | 0.393 |
| Galactica-125M (our reproduction) | 0.417 |
| Galactica-125M (SFT) | 0.512 |
| Chemlactica-125M (SFT) | 0.729 |
| Galactica-125M (SFT - HP tuned) | 0.695 |
| Chemlactica-125M (SFT - HP tuned) | **0.739** |

Table 9: Ablation study on Conditional Generation hyperparameters. Each row represents one combination of Chain-of-Thought (CoT), repetition penalty (rep.), and suppression (supp.). All experiments are done on the molecular weight (top) and SAS (bottom) prediction tasks.

| CoT | rep. | supp. | Chemlactica-125M RMSE (c) ↓ | Invalids ↓ | Chemlactica-1.3B RMSE (c) ↓ | Invalids ↓ | Chemma-2B RMSE (c) ↓ | Invalids ↓ |
|---|---|---|---|---|---|---|---|---|
| No | 1.0 | No | 70.11 (70.11) | 0/100 | 15.81 (65.32) | 1/100 | 12.15 (64.54) | 1/100 |
| Yes | 1.0 | No | 112.52 (112.52) | 0/100 | 187.26 (187.26) | 0/100 | 198.48 (191.89) | 46/100 |
| Yes | 1.010 | No | 82.28 (82.28) | 0/100 | 137.19 (137.19) | 0/100 | 170.02 (170.02) | 0/100 |
| Yes | 1.0 | Yes | 33.46 (33.46) | 0/100 | 18.53 (25.22) | 1/100 | 31.98 (31.85) | 1/100 |
| Yes | 1.005 | Yes | 34.52 (34.52) | 0/100 | 17.14 (17.14) | 0/100 | 29.71 (29.71) | 0/100 |
| **Yes** | **1.010** | **Yes** | **30.27 (30.27)** | **0/100** | **16.87 (16.87)** | **0/100** | **18.93 (20.39)** | **1/100** |
| Yes | 1.015 | Yes | 30.27 (30.27) | 0/100 | 18.07 (19.61) | 1/100 | 18.99 (20.44) | 1/100 |
| Yes | 1.020 | Yes | 31.17 (31.17) | 1/100 | 16.33 (18.03) | 1/100 | 24.16 (25.27) | 1/100 |
| Yes | 1.050 | Yes | 45.38 (45.38) | 1/100 | 16.49 (34.48) | 1/100 | 74.78 (130.11) | 63/100 |
| Yes | 1.100 | Yes | 35.20 (35.20) | 0/100 | 16.61 (32.37) | 1/100 | 740.28 (488.73) | 59/100 |
| No | 1.0 | No | 0.268 (0.769) | 19/100 | 0.395 (0.395) | 1/100 | 0.391 (0.431 | 4/100 |
| Yes | 1.0 | No | 0.887 (0.887) | 0/100 | 0.866 (0.866) | 0/100 | DNF | 46/100 |
| Yes | 1.010 | No | 0.951 (0.951) | 0/100 | 0.691 (0.691) | 0/100 | 0.769 (0.769) | 0/100 |
| Yes | 1.0 | Yes | 0.436 (0.436) | 0/100 | 0.470 (0.470) | 2/100 | 0.253 (0.253) | 1/100 |
| Yes | 1.005 | Yes | 0.439 (0.439) | 0/100 | 0.475 (0.475) | 0/100 | 0.348 (0.363) | 1/100 |
| Yes | 1.010 | Yes | 0.432 (0.432) | 0/100 | **0.281 (0.281)** | **0/100** | **0.275 (0.275)** | **0/100** |
| Yes | 1.015 | Yes | 0.373 (0.378) | 1/100 | 0.540 (0.536) | 2/100 | 0.331 (0.347) | 2/100 |
| Yes | 1.020 | Yes | 0.432 (0.432) | 0/100 | 0.369 (0.369) | 0/100 | 0.325 (0.341) | 2/100 |
| Yes | 1.050 | Yes | 0.294 (0.733) | 3/100 | 0.843 (0.951) | 10/100 | DNF | 63/100 |
| **Yes** | **1.100** | **Yes** | **0.381 (0.381)** | 0/100 | 0.449 (0.449) | 1/100 | DNF | 59/100 |

## A.8 THE ALGORITHM FOR CONVERTING MOLECULES TO PROMPT

Algorithm 2 shows the procedure of converting a molecule and its similar molecule into either a prompt for new molecule generation or a training sample. The separator token represented by <sep> corresponds to the eos token "" for the Chemlactica models and the bos token "<bos>" for Chemma. This keeps consistency with models' training data described in 4.

## A.9 DIVERSITY RESULTS FOR DOCKING MPO; PARP1, FA7, 5HT1B, BRAF, AND JAK2

Table 10 presents the #Circles metric for molecules satisfying the Strict Hit Ratio conditions(Xie et al., 2023). We display the results for our methods alongside the baselines to facilitate comparison. The results demonstrate that our approach generates more diverse high-reward molecules than other methods.

---

**Algorithm 2** molecules2prompt

---

**Input:** $(m_1, m_2, \ldots, m_S), m$
1. Check if the outcome should be a molecule generation prompt or a training sample.
**if** $m$ is $null$ **then**
    1.1. Sample similarity values for molecules in the prompt, desirable oracle score and set the suffix for a molecule generation.
    $v_i^{sim} \sim \mathcal{U}(0.4, 0.9), i = 1, \ldots, S$
    $v^{max} \leftarrow$ the maximum oracle score achieved thus far
    $v^{prop} \sim \mathcal{U}(v^{max}, oracle\_max)$
    $suffix \leftarrow$ [START_SMILES]
**else**
    1.3. Compute the correct similarity values for the molecules in the prompt and the correct oracle score, set the suffix for a training sample.
    $v_i^{sim} = similar(m_i, m), i = 1, \ldots, S$
    $v^{prop} = O(m)$
    $suffix \leftarrow$ [START_SMILES]$m^{smiles}$[END_SMILES]<sep>
**end if**
2. Concatenate all molecules in the prompt with their similarity values.
$p \leftarrow$ [SIMILAR]$m_1^{smiles}$ $v_1^{sim}$[/SIMILAR]$\ldots$[SIMILAR]$m_S^{smiles}$ $v_S^{sim}$[/SIMILAR]
**if** at least one fine-tuning has been performed **then**
    2.1. Add the oracle score to the prompt.
    $p \leftarrow concat(p,$ [PROPERTY]$v^{prop}$[/PROPERTY]$)$
**end if**
3. Add the appropriate suffix.
**return** $concat(p, suffix)$

---

Table 10: Comparison of our approach with other methods. The values represent the #Circles ($\uparrow$) $\pm$ standard deviation for molecules which satisfy the criteria for the Strict Hit Ratio across 10 independent runs. The results of GEAM and Saturn are taken from Lee et al. (2024) and Guo and Schwaller (2024), respectively. Results within one standard deviation from the best one are bolded.

| Method | Target Protein | | | | |
|---|---|---|---|---|---|
| | parp1 | fa7 | 5ht1b | braf | jak2 |
| GEAM | $14 \pm 3$ | $7 \pm 2$ | $25 \pm 3$ | $11 \pm 2$ | $18 \pm 2$ |
| Saturn | $5 \pm 0$ | $3 \pm 1$ | $17 \pm 3$ | $4 \pm 0$ | $7 \pm 1$ |
| Chemlactica-125M | $41 \pm 7$ | $40 \pm 15$ | $38 \pm 7$ | $42 \pm 6$ | $41 \pm 5$ |
| Chemlactica-1.3B | $\mathbf{59 \pm 10}$ | $\mathbf{61 \pm 9}$ | $\mathbf{53 \pm 7}$ | $\mathbf{62 \pm 7}$ | $\mathbf{63 \pm 15}$ |
| Chemma-2B | $\mathbf{66 \pm 16}$ | $\mathbf{73 \pm 14}$ | $\mathbf{64 \pm 14}$ | $\mathbf{72 \pm 14}$ | $\mathbf{72 \pm 19}$ |

## A.10 DETAILED RESULTS FOR PRACTICAL MOLECULAR OPTIMIZATION

Table 11 shows the evaluations of Chemlactica-125M, Chemlactica-1.3B and Gemma-2B, along with other methods on 23 tasks of the PMO benchmark. No method uniformly beats all others on every task. Our (and many other) methods get a zero result on valsartan_smarts. The reason is that the oracle has a binary multiplier term usually equal to zero, so there is no supervision signal for the entire generation process. We separately provide a comparison with MolLEO in Table 12, as the source work did not run the method on all PMO tasks. We present results of our method Chemlactica-125M, Chemlactica-1.3B and Gemma-2B, alongside the MolLEO variants.

Table 11: Comparision of different methods on PMO. The values represent the AUC Top-10 ↑ metric averaged over five independent runs with different seeds.

| Oracle | REINVENT | Augmented Memory | Genetic GFN | Chemlactica 125M | Chemlactica 1.3B | Chemma 2B |
|---|---|---|---|---|---|---|
| albuterol_similarity | 0.882 ± 0.006 | 0.913 ± 0.009 | 0.949 ± 0.010 | **0.951 ± 0.011** | 0.947 ± 0.012 | **0.951 ± 0.009** |
| amlodipine_mpo | 0.635 ± 0.035 | 0.691 ± 0.047 | 0.761 ± 0.019 | **0.772 ± 0.091** | 0.769 ± 0.083 | 0.766 ± 0.107 |
| celecoxib_rediscover | 0.713 ± 0.067 | 0.796 ± 0.008 | 0.802 ± 0.029 | 0.906 ± 0.046 | 0.911 ± 0.013 | **0.920 ± 0.011** |
| deco_hop | 0.666 ± 0.044 | 0.658 ± 0.024 | 0.733 ± 0.109 | 0.801 ± 0.101 | **0.836 ± 0.117** | 0.831 ± 0.123 |
| drd2 | 0.945 ± 0.007 | 0.963 ± 0.006 | **0.974 ± 0.006** | 0.965 ± 0.007 | 0.968 ± 0.005 | 0.972 ± 0.006 |
| fexofenadine_mpo | 0.784 ± 0.006 | 0.859 ± 0.009 | 0.856 ± 0.039 | 0.881 ± 0.031 | 0.891 ± 0.039 | **0.931 ± 0.014** |
| gsk3 | 0.865 ± 0.043 | 0.881 ± 0.021 | 0.881 ± 0.042 | 0.926 ± 0.022 | 0.916 ± 0.027 | **0.928 ± 0.021** |
| isomers_c7h8n2o2 | 0.852 ± 0.036 | 0.853 ± 0.087 | 0.969 ± 0.003 | **0.951 ± 0.012** | 0.933 ± 0.017 | 0.947 ± 0.009 |
| isomers_c9h10n2o2pf2cl | 0.642 ± 0.054 | 0.736 ± 0.051 | 0.897 ± 0.007 | 0.927 ± 0.006 | **0.929 ± 0.012** | 0.914 ± 0.017 |
| jnk3 | 0.783 ± 0.023 | 0.739 ± 0.110 | 0.764 ± 0.069 | 0.881 ± 0.058 | 0.866 ± 0.021 | **0.891 ± 0.032** |
| median1 | 0.356 ± 0.009 | 0.326 ± 0.013 | 0.379 ± 0.010 | 0.359 ± 0.060 | **0.382 ± 0.047** | **0.382 ± 0.022** |
| median2 | 0.276 ± 0.008 | 0.291 ± 0.008 | 0.294 ± 0.007 | 0.328 ± 0.032 | 0.329 ± 0.016 | **0.366 ± 0.018** |
| mestranol_similarity | 0.618 ± 0.048 | 0.750 ± 0.049 | 0.708 ± 0.057 | 0.896 ± 0.064 | 0.850 ± 0.051 | **0.926 ± 0.023** |
| osimertinib_mpo | 0.837 ± 0.009 | 0.855 ± 0.004 | 0.860 ± 0.008 | **0.907 ± 0.015** | 0.892 ± 0.013 | 0.879 ± 0.016 |
| perindopril_mpo | 0.537 ± 0.016 | 0.613 ± 0.015 | 0.595 ± 0.014 | 0.709 ± 0.052 | **0.755 ± 0.066** | 0.711 ± 0.062 |
| qed | 0.941 ± 0.000 | **0.942 ± 0.000** | **0.942 ± 0.000** | **0.942 ± 0.000** | **0.942 ± 0.000** | 0.941 ± 0.000 |
| ranolazine_mpo | 0.760 ± 0.009 | 0.801 ± 0.006 | 0.819 ± 0.018 | 0.864 ± 0.014 | **0.883 ± 0.017** | 0.868 ± 0.015 |
| scaffold_hop | 0.560 ± 0.019 | 0.567 ± 0.008 | 0.615 ± 0.100 | 0.626 ± 0.016 | **0.673 ± 0.080** | 0.669 ± 0.110 |
| sitagliptin_mpo | 0.021 ± 0.003 | 0.284 ± 0.050 | 0.634 ± 0.039 | **0.649 ± 0.051** | 0.586 ± 0.062 | 0.613 ± 0.018 |
| thiothixene_rediscovery | 0.534 ± 0.013 | 0.550 ± 0.041 | 0.583 ± 0.034 | 0.624 ± 0.102 | 0.693 ± 0.119 | **0.698 ± 0.121** |
| troglitazone_rediscovery | 0.441 ± 0.032 | 0.540 ± 0.048 | 0.511 ± 0.054 | 0.734 ± 0.130 | 0.765 ± 0.138 | **0.824 ± 0.049** |
| valsartan_smarts | **0.178 ± 0.358** | 0.000 ± 0.000 | 0.135 ± 0.271 | 0.000 ± 0.000 | 0.000 ± 0.000 | 0.000 ± 0.000 |
| zaleplon_mpo | 0.358 ± 0.062 | 0.394 ± 0.026 | 0.552 ± 0.033 | 0.569 ± 0.047 | 0.569 ± 0.020 | **0.608 ± 0.055** |
| sum | 14.196 | 15.002 | 16.213 | 17.170 ± 0.424 | 17.284 ± 0.284 | **17.534 ± 0.214** |

Table 12: Comparision with MolLEO variants on PMO tasks reported in the source work (Wang et al., 2024). The values represent the AUC Top-10 ↑ metric averaged over five independent runs with different seeds.

| Method | MolLEO (MolSTM) | MolLEO (MolT5) | MolLEO (GPT-4) | Chemlactica 125M | Chemlactica 1.3B | Chemma 2B |
|---|---|---|---|---|---|---|
| QED | 0.937 ± 0.002 | 0.937 ± 0.002 | **0.948 ± 0.004** | 0.942 ± 0.000 | 0.942 ± 0.000 | 0.941 ± 0.000 |
| JNK3 | 0.643 ± 0.226 | 0.728 ± 0.079 | 0.790 ± 0.027 | 0.881 ± 0.058 | 0.866 ± 0.021 | **0.891 ± 0.032** |
| DRD2 | 0.975 ± 0.003 | **0.981 ± 0.002** | 0.968 ± 0.012 | 0.965 ± 0.007 | 0.968 ± 0.005 | 0.972 ± 0.006 |
| GSK3$\beta$ | 0.898 ± 0.041 | 0.889 ± 0.015 | 0.863 ± 0.047 | 0.926 ± 0.022 | 0.916 ± 0.027 | **0.928 ± 0.021** |
| mestranol_similarity | 0.596 ± 0.018 | 0.717 ± 0.104 | **0.972 ± 0.009** | 0.896 ± 0.064 | 0.850 ± 0.051 | 0.926 ± 0.023 |
| thiothixene_rediscovery | 0.508 ± 0.035 | 0.696 ± 0.081 | **0.727 ± 0.052** | 0.624 ± 0.102 | 0.693 ± 0.119 | 0.698 ± 0.121 |
| perindopril_mpo | 0.554 ± 0.037 | 0.738 ± 0.016 | 0.600 ± 0.031 | 0.709 ± 0.052 | **0.755 ± 0.066** | 0.711 ± 0.062 |
| ranolazine_mpo | 0.725 ± 0.040 | 0.749 ± 0.012 | 0.769 ± 0.022 | 0.864 ± 0.014 | **0.883 ± 0.017** | 0.868 ± 0.015 |
| sitagliptin_mpo | 0.548 ± 0.065 | 0.506 ± 0.100 | 0.584 ± 0.067 | **0.649 ± 0.051** | 0.586 ± 0.062 | 0.613 ± 0.018 |
| isomers_c9h10n2o2pf2cl | 0.871 ± 0.039 | 0.873 ± 0.019 | 0.874 ± 0.053 | 0.927 ± 0.006 | **0.929 ± 0.012** | 0.914 ± 0.017 |
| deco_hop | 0.613 ± 0.016 | 0.827 ± 0.093 | **0.942 ± 0.013** | 0.801 ± 0.101 | 0.836 ± 0.117 | 0.831 ± 0.123 |
| scaffold_hop | 0.527 ± 0.019 | 0.559 ± 0.102 | **0.971 ± 0.004** | 0.626 ± 0.016 | 0.673 ± 0.080 | 0.669 ± 0.110 |
| sum | 8.395 | 9.202 | 10.008 | 9.81 | 9.893 | 9.962 |

## A.11 Analysis of Molecular Optimization

### A.11.1 Ablation on Finetuning During Optimization

A key component of our proposed optimization algorithm is the fine-tuning step, initiated when the algorithm's progress stagnates. To assess the impact of this fine-tuning step, we conducted a comparative analysis of optimization processes both with and without this feature. For this evaluation, we selected four representative tasks from the PMO benchmark: jnk3, median1, sitagliptin_mpo, and scaffold_hop. We select these tasks to provide diverse challenges and adequately represent the full suite of PMO tasks.

Table 13 presents the quantitative results of these experiments. To provide a more comprehensive understanding of the fine-tuning effect, we visualize the optimization trajectories in Figures 8 through 10. These visualizations aggregate data from five independent runs, offering insights into both the mean performance and its variance across different initializations.

This ablation study allows us to isolate the impact of the fine-tuning step and understand its contribution to the overall performance of our optimization algorithm across different types of molecular optimization tasks.

Table 13: Illustration of the results of ablation study on the fine-tuning step in the optimization algorithm. The values represent AUC Top-10 $\uparrow$ obtained from five independent runs.

| | Chemlactica-125M | | Chemlactica-1.3B | | Chemma-2B | |
|---|---|---|---|---|---|---|
| | fine-tuning | no fine-tuning | fine-tuning | no fine-tuning | fine-tuning | no fine-tuning |
| jnk3 | $0.881 \pm 0.058$ | $0.878 \pm 0.040$ | $0.866 \pm 0.021$ | $0.867 \pm 0.036$ | $0.891 \pm 0.032$ | $0.869 \pm 0.033$ |
| median1 | $0.359 \pm 0.060$ | $0.371 \pm 0.006$ | $0.382 \pm 0.047$ | $0.395 \pm 0.027$ | $0.382 \pm 0.022$ | $0.380 \pm 0.034$ |
| scaffold_hop | $0.626 \pm 0.016$ | $0.648 \pm 0.017$ | $0.673 \pm 0.080$ | $0.721 \pm 0.121$ | $0.669 \pm 0.110$ | $0.700 \pm 0.122$ |
| sitagliptin_mpo | $0.649 \pm 0.051$ | $0.607 \pm 0.051$ | $0.586 \pm 0.062$ | $0.576 \pm 0.082$ | $0.613 \pm 0.018$ | $0.563 \pm 0.059$ |
| sum | $\mathbf{2.515 \pm 0.119}$ | $2.504 \pm 0.068$ | $2.506 \pm 0.155$ | $\mathbf{2.559 \pm 0.062}$ | $\mathbf{2.555 \pm 0.099}$ | $2.512 \pm 0.160$ |

Table 14: The performance of the extended version of our optimization algorithm on selected PMO tasks. The prompts used in the optimization contain the description of the tasks in the format our language models has seen during pretraining. See Table 15 for the additional tags used in the prompts.

| | Chemlactica-125M | | Chemlactica-1.3B | | Chemma-2B | |
|---|---|---|---|---|---|---|
| | no add. props. | add. props. | no add. props. | add. props. | no add. props. | add. props. |
| jnk3 | $0.881 \pm 0.058$ | $0.881 \pm 0.058$ | $0.866 \pm 0.021$ | $0.866 \pm 0.021$ | $0.891 \pm 0.032$ | $0.891 \pm 0.032$ |
| median1 | $0.359 \pm 0.060$ | $0.479 \pm 0.004$ | $0.382 \pm 0.047$ | $0.488 \pm 0.000$ | $0.382 \pm 0.022$ | $0.479 \pm 0.002$ |
| scaffold_hop | $0.626 \pm 0.016$ | $0.983 \pm 0.004$ | $0.673 \pm 0.080$ | $0.975 \pm 0.006$ | $0.669 \pm 0.110$ | $0.983 \pm 0.003$ |
| sitagliptin_mpo | $0.649 \pm 0.051$ | $0.534 \pm 0.041$ | $0.586 \pm 0.062$ | $0.495 \pm 0.035$ | $0.613 \pm 0.018$ | $0.576 \pm 0.055$ |
| sum | $2.515 \pm 0.119$ | $\mathbf{2.920 \pm 0.096}$ | $2.506 \pm 0.155$ | $\mathbf{2.824 \pm 0.034}$ | $2.555 \pm 0.099$ | $\mathbf{2.887 \pm 0.040}$ |

### A.11.2 LEVERAGING KNOWN MOLECULAR PROPERTIES IN OPTIMIZATION TASKS

Our language models possess knowledge of various molecular properties such as QED, CLogP, and TPSA. However, we deliberately avoid utilizing this information in Algorithm 1 to maintain fair comparison with other methods. This decision stems from the fact that our models have been trained on properties that are components of the oracle functions we optimize against (e.g., those in PMO). Exploiting this partial oracle information could potentially give our method an unfair advantage.

We conducted a separate set of experiments to explore the models' capacity to utilize additional information in solving optimization problems using four tasks from the PMO benchmark: jnk3, median1, sitagliptin_mpo, and scaffold_hop. For these tasks, we modified Algorithm 2 to incorporate relevant known properties into the prompt $p$ between steps 2 and 3.

Table 14 presents a performance comparison between our standard approach and this property-augmented version. The specific syntax used for adding these properties to the prompts is detailed in Table 15. Notably, no additional properties were added for the jnk3 docking function as our models lack specific knowledge about this component.

The results demonstrate a significant performance improvement across all models when these additional properties are incorporated. This finding suggests that our models can effectively leverage their pre-existing knowledge of molecular properties to enhance their performance in molecular design tasks. However, it is important to note that while this approach showcases the potential of our models, it may not provide a fair comparison with methods that do not have access to such property information.

Table 15: The descriptions of tasks used in the prompts in the extended version of our optimization algorithm. The results are in Table 14. See Section A.11.2 for details.

| | the syntax of additional properties added to the prompts |
|---|---|
| jnk3 | (nothing added) |
| median1 | [SIMILAR]$camphor\_smiles$ 0.55[/SIMILAR][SIMILAR]$menthol\_smiles$ 0.55[/SIMILAR] |
| scaffold_hop | [SIMILAR]$pharmacophor\_smiles$ 0.80[/SIMILAR] |
| sitagliptin_mpo | [SIMILAR]$sitagliptin\_smiles$ 0.99[/SIMILAR][CLOGP]2.02[/CLOGP][TPSA]77.04[/TPSA] |

Table 16: Impact of numerical precision on Docking MPO experiments from 6.2.2. Oracle burden and generative yield values are reward-threshold dependent. The numbers next to the metrics correspond to the thresholds, and the values in parentheses for oracle burden indicate how many unique molecules need to be generated. The best performance on each task-metric combination is bolded. We use the best-performing hyperparameters from the PMO benchmark.

| Metric | Target | Chemlactica-125M BF16 | Chemlactica-125M FP32 |
|---|---|---|---|
| Generative Yield 0.7 ↑ | DRD2 | $3501 \pm 252$ | $\mathbf{3733 \pm 512}$ |
| | MK2 | $3000 \pm 80$ | $\mathbf{3772 \pm 578}$ |
| | AChE | $\mathbf{4337 \pm 133}$ | $4108 \pm 67$ |
| Generative Yield 0.8 ↑ | DRD2 | $2574 \pm 103$ | $\mathbf{2827 \pm 510}$ |
| | MK2 | $1223 \pm 519$ | $\mathbf{2569 \pm 1156}$ |
| | AChE | $\mathbf{3877 \pm 272}$ | $3246 \pm 168$ |
| Oracle burden 0.8 (1) ↓ | DRD2 | $156 \pm 100$ | $\mathbf{20 \pm 29}$ |
| | MK2 | $\mathbf{320 \pm 83}$ | $345 \pm 312$ |
| | AChE | $\mathbf{10 \pm 8}$ | $22 \pm 28$ |
| Oracle burden 0.8 (10) ↓ | DRD2 | $283 \pm 61$ | $\mathbf{114 \pm 08}$ |
| | MK2 | $631 \pm 100$ | $\mathbf{493 \pm 418}$ |
| | AChE | $\mathbf{123 \pm 119}$ | $224 \pm 17$ |
| Oracle burden 0.8 (100) ↓ | DRD2 | $577 \pm 71$ | $\mathbf{364 \pm 119}$ |
| | MK2 | $1134 \pm 178$ | $\mathbf{865 \pm 533}$ |
| | AChE | $\mathbf{350 \pm 137}$ | $497 \pm 58$ |

### A.11.3 THE IMPACT OF FLOATING POINT PRECISION ON MOLECULAR OPTIMIZATION

**Numerical Precision in Model Training**    Lower precision training, including mixed and half-precision methods, is commonly used to increase training throughput. These techniques, employed during our models' pretraining stages, typically have negligible impact on performance and may even provide a regularizing effect(Micikevicius et al., 2017). However, in molecular optimization involving multiple rounds of fine-tuning, lower numerical precision leads to significantly degraded performance. Several factors contribute to this phenomenon in the specific case of molecular optimization with language models.

**Challenges in Batched Generation**    Molecular optimization pipelines require repeated model calls for generation, followed by oracle function scoring. While batched processing accelerates this process through GPU parallelization, it introduces complications. The necessary padding for batch processing alters matrix sizes, affecting multiply-accumulate operations within the model. These small errors accumulate as they propagate through the model's layers. Lower precision exacerbates these errors, leading to larger discrepancies in logit values and, consequently, more significant impacts on the generated molecules.

**Cascading Effects of Sub-optimal Generations**    In our approach, high-scoring generated molecules are leveraged for fine-tuning and generating similar structures that steer the optimization processs. Thus, when lower precision leads to sub-optimal molecule generation, it creates a positive feedback loop. The model is fine-tuned on and guided by these lower-quality molecules, hindering the generation of higher-scoring molecules in subsequent iterations. This causal relationship between successive generations underlies the adverse effects of low-precision training and inference in molecular optimization pipelines.

**Precision Ablation Study**    To quantify the impact of numerical precision on the optimization process, we conducted an ablation study comparing 32-bit floating point precision with bfloat16 precision. Table 16 presents the results of this comparison across all drug discovery case studies described in Section 6.2.2. We show that for the majority of task-metric combinations, optimization results were better when model parameters were in full floating point precision. Despite the potential computational costs, these results demonstrate the importance of maintaining higher numerical precision in molecular optimization tasks.

### A.12 ADDITIONAL EXPERIMENTS

### A.12.1 QED MAXIMIZATION WITH SIMILARITY CONSTRAINED MOLECULAR DESIGN

**Problem formulation.**    This optimization problem aims to generate a molecule with a high QED, similar to another given molecule. More formally, given a molecule $M$, the objective is to generate a new molecule $M'$ such that $sim(M', M) \geq 0.4$ and $qed(M') \geq 0.9$. Following Wang et al. (2023), 800 molecules are selected with QED in the range $[0.7, 0.8]$ as the inputs to the optimization problem, and the measure of performance is the percentage of the molecules that have been optimized (satisfy the QED and similarity constraints). In addition, a maximum number of QED evaluations is chosen to optimize each lead molecule.

**Our approach.** Since this is a lead optimization problem, we add the lead molecule to all prompts in addition to the molecules added from the pool. The lead molecule is added by enclosing it in `[SIMILAR]` tag. For this task, we design an oracle function by combining the QED value of the generated molecule with the similarity value of the lead molecule with the generated molecule. Additionally, we decreased the maximum number of QED evaluations to 10000, compared to the baselines, which used 50000.

**Results.** For this task, we only evaluate the Chemlactica-125M model, which achieves better success rates compared to the best existing approaches, $99.0\%$ (Chemlactica-125M) versus $94.6\%$ (RetMol), while being constrained to use five times less QED evaluations at maximum. Since the performance of the Chemlactica-125M saturates the benchmark, we have not evaluated other models for this task. Table 17 illustrates the performance of different algorithms.

### A.12.2 PROPERTY PREDICTION

**Supervised fine-tuning recipe.** Inspired by instruction tuning methodologies and Zhou et al. (2023), we generated a specialized training corpus formatted as follows:

`[START_SMILES]`$m^{smiles}$`[END_SMILES][PROPERTY]`activity `<VALUE>[/PROPERTY]`.

**Hyperparameters.** We only trained the model on generated responses following the [PROPERTY] tag during the fine-tuning process. Our initial experiments indicated that a general fine-tuning recipe of 15 epochs yielded satisfactory results with a peak learning rate of $10e-4$, 3 epochs of warmup and a NEFTune noise (Jain et al., 2023) of 5. To further improve model performance, we conducted an extensive hyperparameter tuning study, exploring a grid of values within the following ranges: Learning rate: [0.00001, 0.00005, 0.0001, 0.0002], Number of epochs: [10, 15, 20], Warmup epochs: [0, 1, 2, 3], NEFTune noise : [0.0, 5.0, 10.0]. In our experiments, we employed a batch size of 32 and a maximum sequence length of 128, except in cases where GPU memory limitations necessitated reducing the batch size to 16 while maintaining the established sequence length. Table 18 shows the best values for all tasks and models.

**Results.** Table 2 lists the results for three regression tasks from MoleculeNet (Wu et al., 2018) alongside other comparable methods like Chilingaryan et al. (2024) and Ross et al. (2021). For all Moleculenet tasks, we have utilized the DeepChem library Ramsundar et al. (2019) and the original recommended splits to load the datasets. .Fang et al. (2023b) introduces a novel dataset encompassing six ADME targets. The assessment of ADME properties is crucial for understanding how potential drug candidates interact with the human body, aspects of which are absorption, distribution, metabolism, and excretion. This knowledge is essential for evaluating efficacy, safety, and clinical potential, guiding drug development toward optimal therapeutic outcomes. The authors have disclosed DMPK datasets collected over a 20-month period, focusing on six ADME in vitro endpoints: human and rat liver microsomal stability, MDR1-MDCK efflux ratio, solubility, and human and rat plasma protein binding. The dataset comprises between 885 and 3087 measurements for each corresponding endpoint. For this series of tasks, we utilized Polaris Hub Wognum et al. (2024) as a centralized platform for dataset loading and result sharing. To promote standardized benchmarks in the field, we employed the datasets as presented, with default preprocessing. We limited our comparisons to the results available at the time, including the baselines provided by the original authors. We generated a randomly split validation set for this series of tasks, comprising 20 percent of the training data. After identifying the optimal hyperparameters, we trained on the entire training set to maximize performance. Table 19 presents the results for ADME tasks. The presented results showcase the abilities of our models after the hyperparameter tuning stage.

### A.12.3 MODEL CALIBRATION

**Methodology** Model calibration in language modeling refers to the alignment between a model's predicted probabilities for generating specific text and the actual likelihood of that text being correct. To assess the calibration of our models, we developed a suite of multiple-choice property prediction questions based on our training data format.

Table 17: Performance comparison of different algorithms on QED and Similarity constrained molecular optimization problem.

| | Success Rate (%) ↑ |
|---|---|
| QMO | 92.8 |
| RetMol | 94.5 |
| Chemlactica-125M | **99.0** |

We generated 2000 questions for each computed property, resulting in 10,000 responses. Each question presented a SMILES string as input:

`[START_SMILES]`$m^{smiles}$`[END_SMILES]`

and was followed by five potential continuations, with only one being correct. An example of such a continuation for a question testing QED could be: `[QED]`$0.78$`[/QED]`.

This methodology is inspired by the calibration analysis in the GPT-4 technical report (OpenAI, 2023), which highlights calibration as a key indicator of high-quality pretraining. For each response, we calculated the model's predicted probability from the perplexity of the text, normalizing it against other responses for the same question. These probabilities were then aggregated and sorted into 10 equal-width bins. We plotted the fraction of correct responses for each bin, allowing us to visualize the relationship between the model's confidence and accuracy.

**Results**  Figures 3a and 3b present the calibration plots for Chemma-2B and Chemlactica-125M, respectively. The x-axis represents the 10 probability bins, while the left y-axis shows the correct response fraction. The right y-axis and red bars indicate the number of occurrences within each bin.

Chemlactica and Chemma models demonstrate robust calibration, as evidenced by the near-linear relationship between assigned probabilities and correct outcomes across all computed properties. This relationship closely follows the diagonal grey line, which represents perfect calibration.

These results suggest that the perplexity scores generated by our models serve as reliable confidence indicators for molecular data predictions (averaged over a set of molecules), provided the data falls within the distribution of the training corpus. This calibration is crucial for practical applications, as it allows users to accurately gauge the reliability of the models' outputs in simple molecular prediction and generation tasks. However, finetuning, like that performed in the optimization algorithm, likely leads to a loss of model calibration(OpenAI, 2023).

### A.13  ADDITIONAL FIGURES

#### A.13.1  VISUALIZATION OF THE MODEL OUTPUTS ON PROPERTY PREDICTION AND CONDITIONAL GENERATION TASKS

Figures 4e-4e show the performance of Chemma-2B for property prediction and conditional molecular generation tasks. Each dot in the scatter plot corresponds to one molecule. The histogram in the background is the distribution of those properties in our training set. The purple line represents the RMSE between the property's ground truth and predicted values.

Table 18: Selected hyperparameters for property prediction tasks as a result of the grid search. We report learning rate (LR), warmup ratio (WU), number of epochs (Ep.) and Neftune noise (Nef.).

| Task | Chemlactica-125M | | | | Chemlactica-1B | | | | Chemma-2B | | | |
|---|---|---|---|---|---|---|---|---|---|---|---|---|
| | LR | WU | Ep. | Nef. | LR | WU | Ep. | Nef. | LR | WU | Ep. | Nef. |
| RLM | 5.0e-5 | 0 | 15 | 0 | 1.0e-5 | 3 | 10 | 5 | 2.0e-4 | 1 | 20 | 10 |
| HLM | 5.0e-5 | 1 | 10 | 5 | 1.0e-5 | 3 | 10 | 0 | 1.0e-4 | 3 | 20 | 0 |
| MDR | 5.0e-5 | 2 | 20 | 0 | 1.0e-5 | 1 | 15 | 0 | 2.0e-4 | 1 | 20 | 5 |
| RPPB | 1.0e-4 | 1 | 15 | 0 | 1.0e-5 | 3 | 10 | 0 | 1.0e-4 | 1 | 15 | 5 |
| HPPB | 2.0e-4 | 3 | 20 | 10 | 1.0e-4 | 2 | 15 | 10 | 2.0e-4 | 2 | 15 | 0 |
| SOL | 1.0e-4 | 2 | 10 | 0 | 5.0e-5 | 0 | 20 | 5 | 2.0e-4 | 1 | 15 | 10 |
| FREESOLV | 1.0e-4 | 0 | 15 | 5 | 1.0e-5 | 3 | 10 | 10 | 1.0e-4 | 2 | 15 | 0 |
| ESOL | 1.0e-4 | 0 | 10 | 0 | 1.0e-5 | 3 | 10 | 5 | 2.0e-4 | 3 | 10 | 0 |
| LIPO | 5.0e-5 | 1 | 2 | 0 | 1.0e-5 | 0 | 10 | 0 | 2.0e-4 | 2 | 10 | 0 |

Table 19: Regression tasks from the ADME benchmark. All numbers are Pearson correlation ↑. RandomForestRegressors use task-specific features: [a] desc2D, [b] fcfp4, [c] atompair.

| | HLM | MDR | SOL | RLM | HPPB | RPPB |
|---|---|---|---|---|---|---|
| 1B_MPNN (LargeMix-and-Phenomics) | **0.778** | **0.860** | **0.764** | **0.784** | **0.888** | **0.908** |
| adme-fang-RandomForestRegressor | $0.639^a$ | $0.716^a$ | $0.439^b$ | $0.640^a$ | $0.690^c$ | $0.722^a$ |
| Chemlactica-125M | 0.717 | 0.714 | 0.608 | 0.714 | 0.774 | 0.442 |
| Chemlactica-1.3B | 0.720 | 0.762 | 0.574 | 0.698 | 0.635 | 0.614 |
| Chemma-2B | 0.674 | 0.709 | 0.558 | 0.660 | 0.636 | 0.747 |

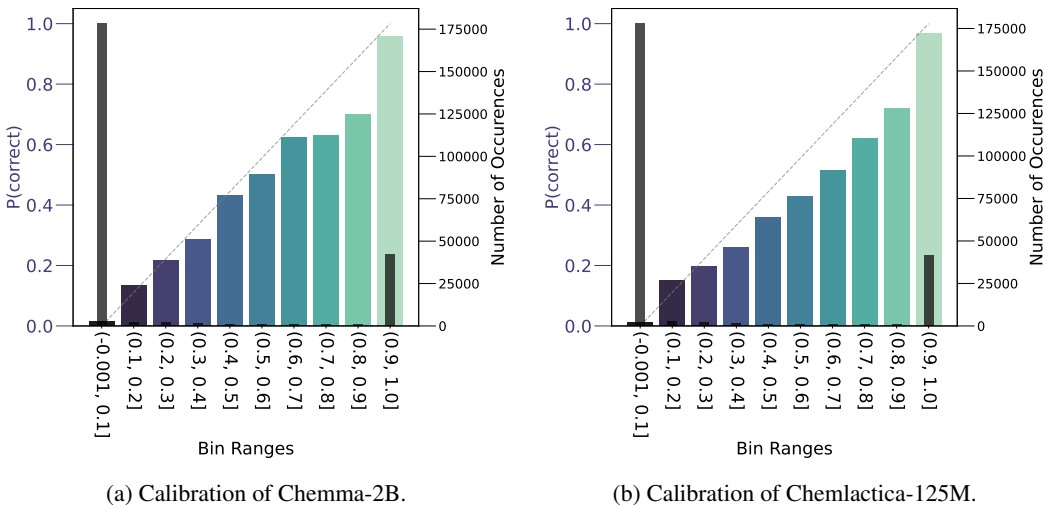

(a) Calibration of Chemma-2B.          (b) Calibration of Chemlactica-125M.

Figure 3: Model calibration on synthetic multiple choice question where y=x represents perfect calibration.

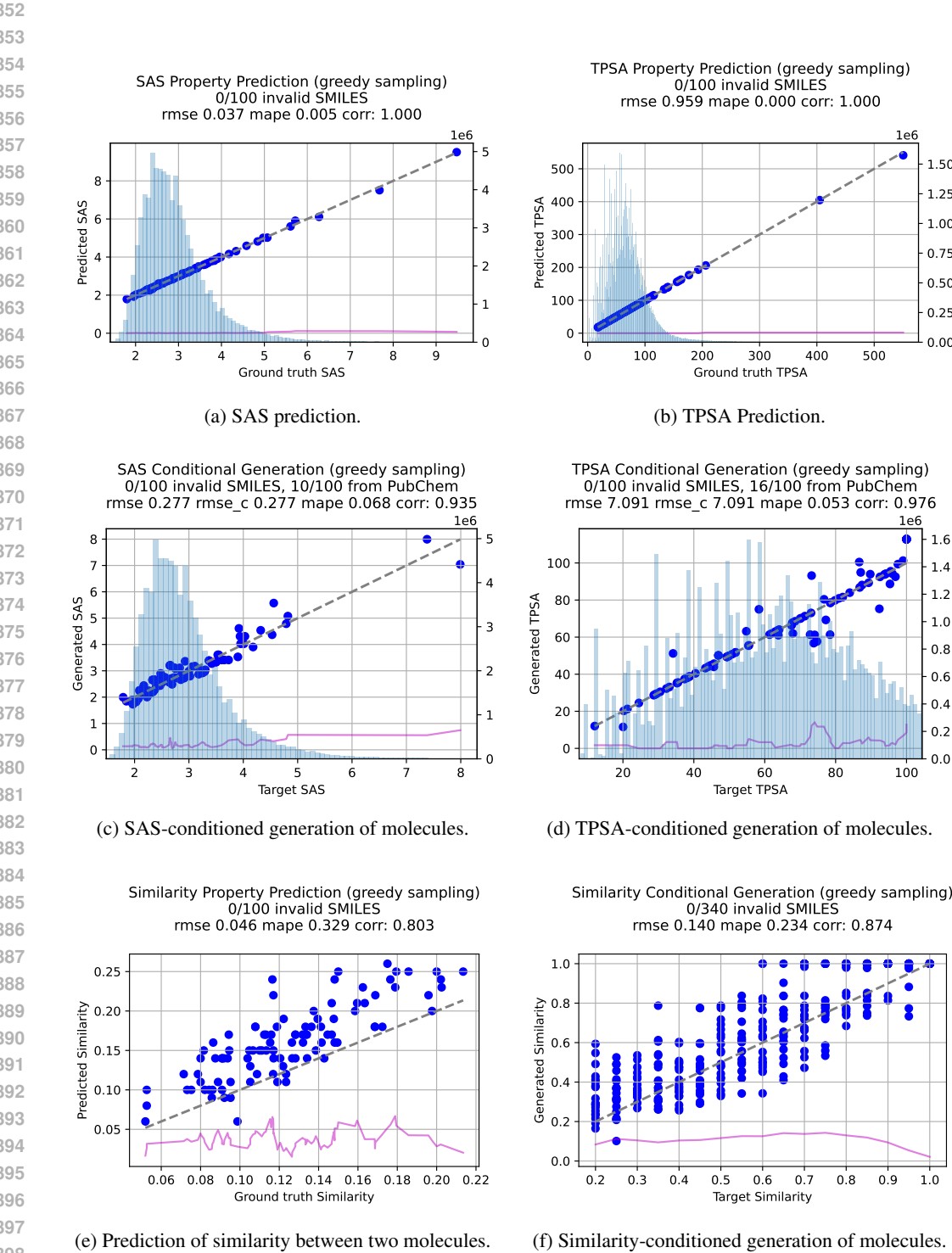

(a) SAS prediction.

(b) TPSA Prediction.

(c) SAS-conditioned generation of molecules.

(d) TPSA-conditioned generation of molecules.

(e) Prediction of similarity between two molecules.

(f) Similarity-conditioned generation of molecules.

Figure 4: Illustration of errors made by Chemma-2B during property prediction and conditional generation for various properties.

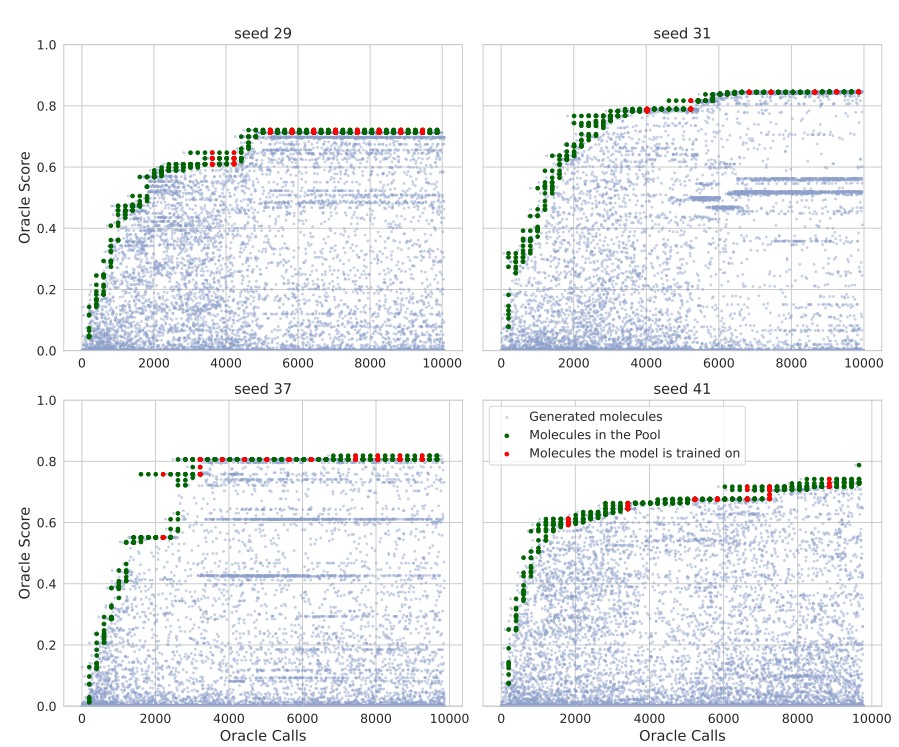

Figure 5: Optimization process visualization using Chemlactica-125M model for `sitagliptin_mpo` task with four different seeds.

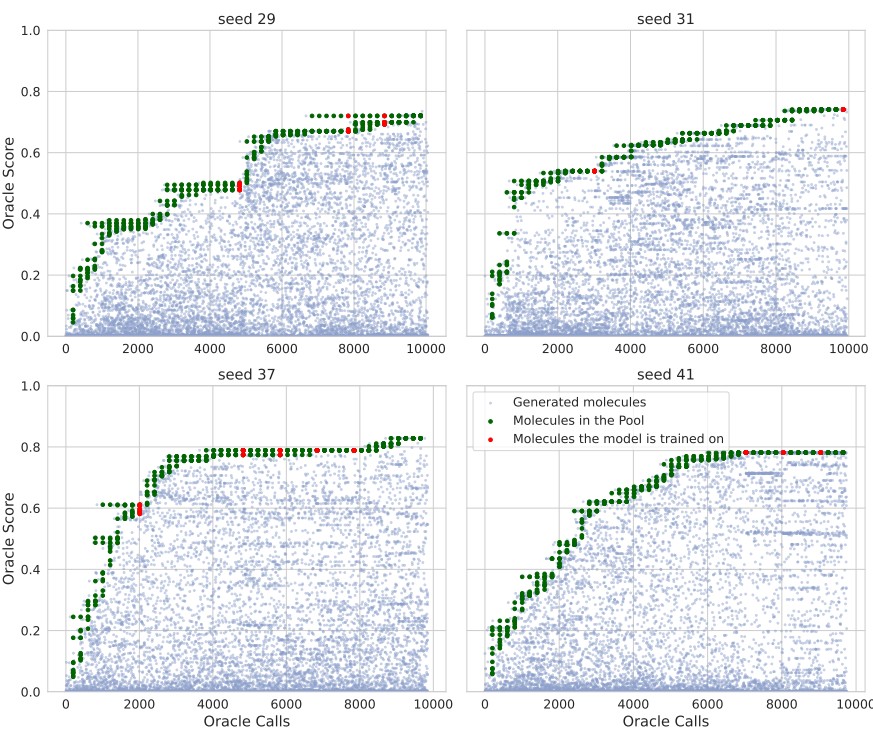

Figure 6: Optimization process visualization using Chemlactica-1.3B model for `sitagliptin_mpo` task with four different seeds.

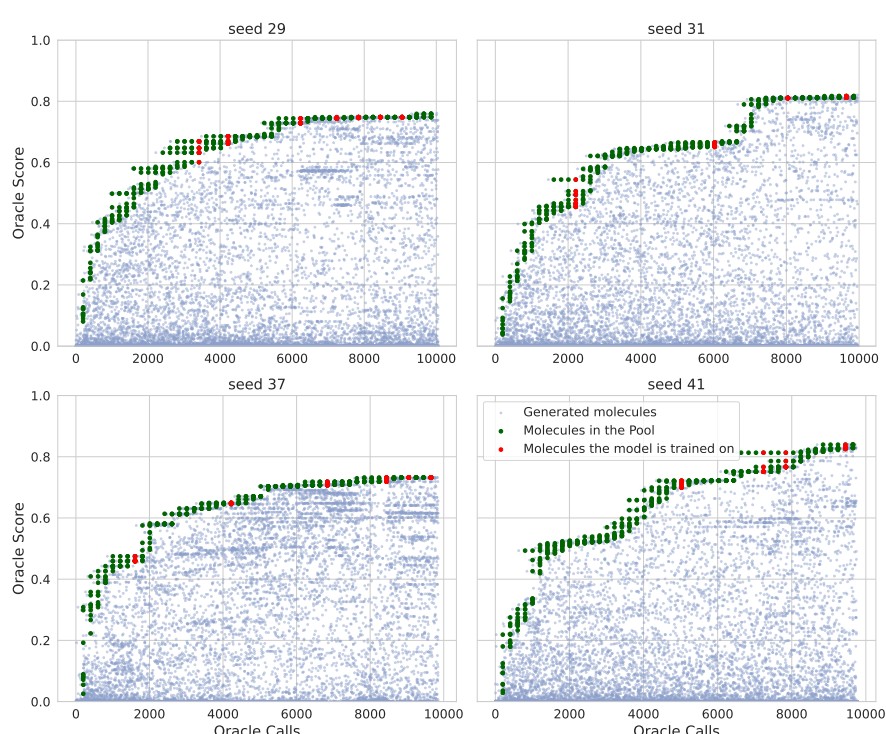

Figure 7: Optimization process visualization using Chemma-2B model for `sitagliptin_mpo` task with four different seeds.

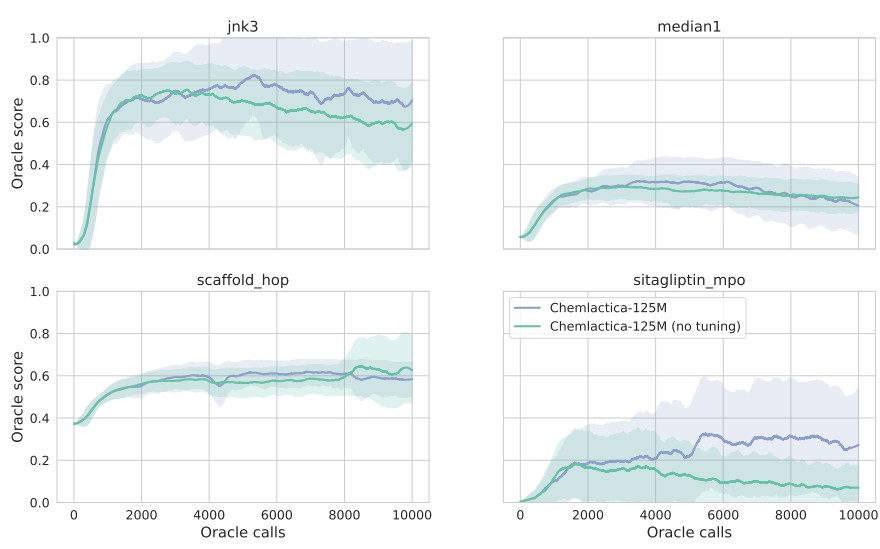

Figure 8: Mean oracle score ± standard deviation of the generated molecule for Chemlactica-125M.

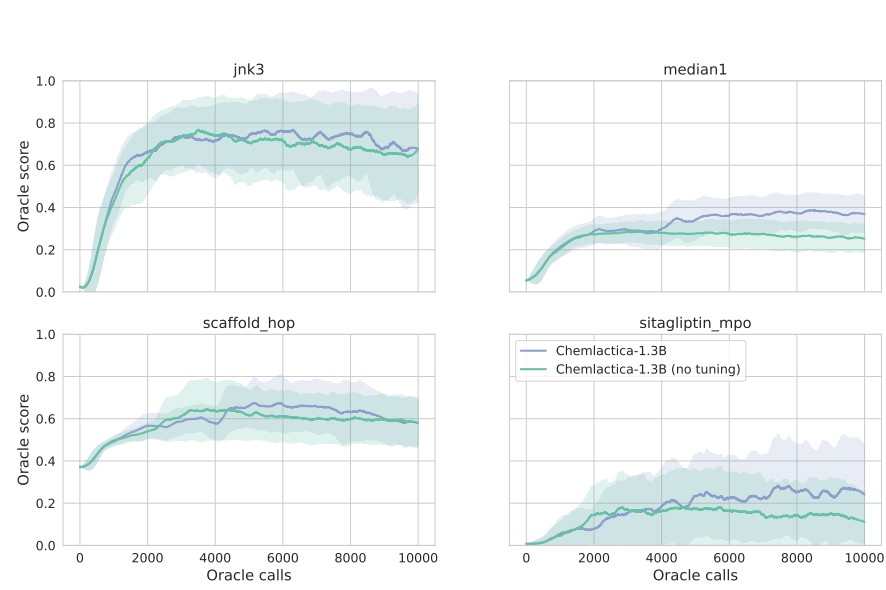

Figure 9: Mean oracle score $\pm$ standard deviation of the generated molecule for Chemlactica-1.3B.

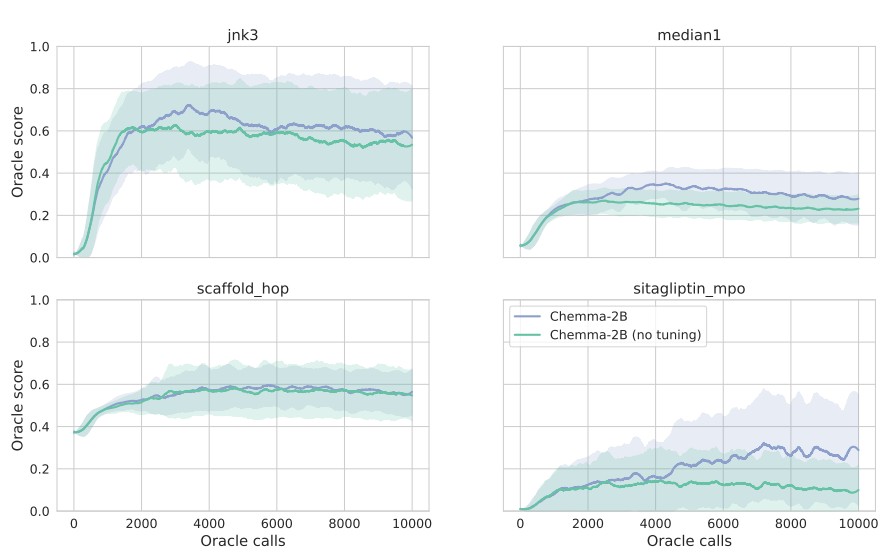

Figure 10: Mean oracle score $\pm$ standard deviation of the generated molecule for Chemma-2B.

### A.13.2 Docking Scores Throughout DRD2 MPO

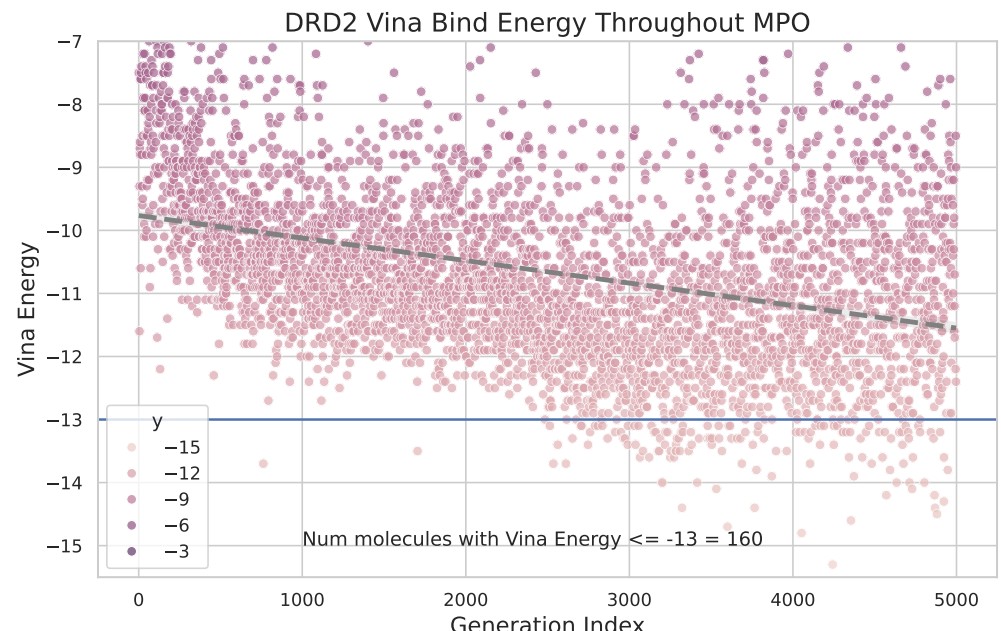

Figure 11: Autodock Vina energy scores for molecules generated through DRD2 MPO process; note that molecules which did not dock at all are not included.

### A.13.3 Generated Molecules from the Docking Experiments

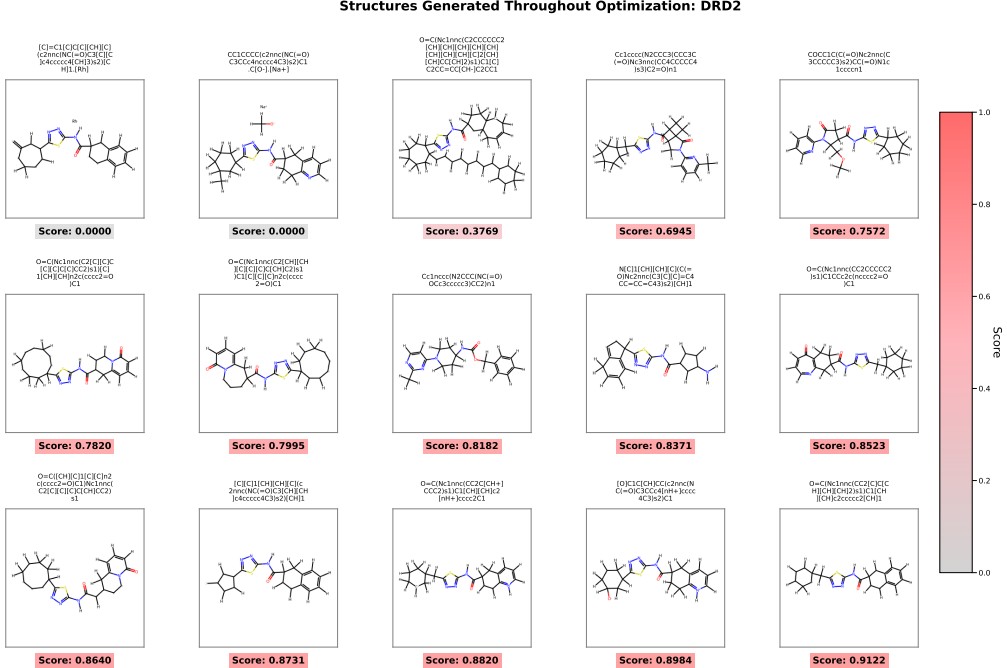

**Structures Generated Throughout Optimization: MK2**

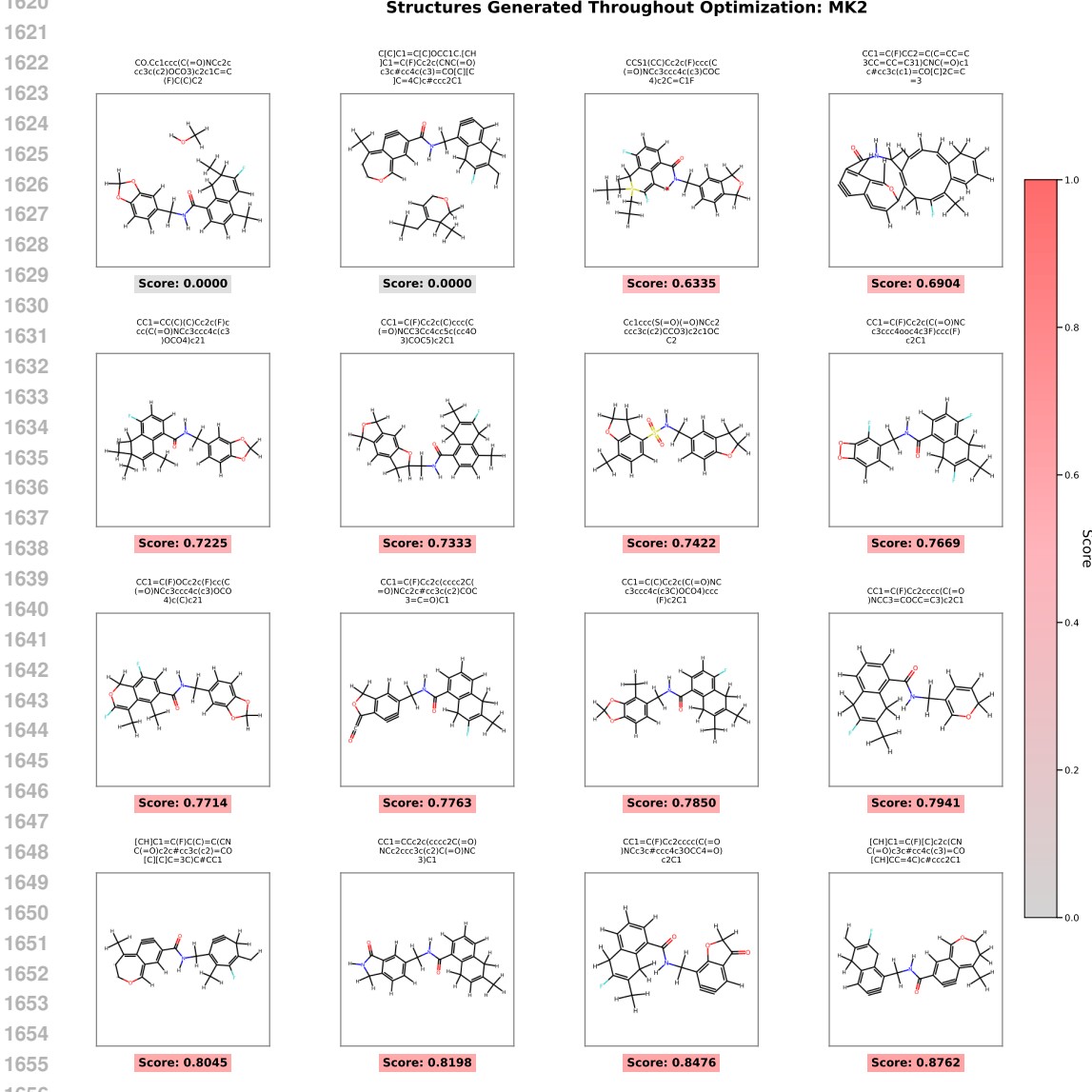

**Structures Generated Throughout Optimization: AChE**

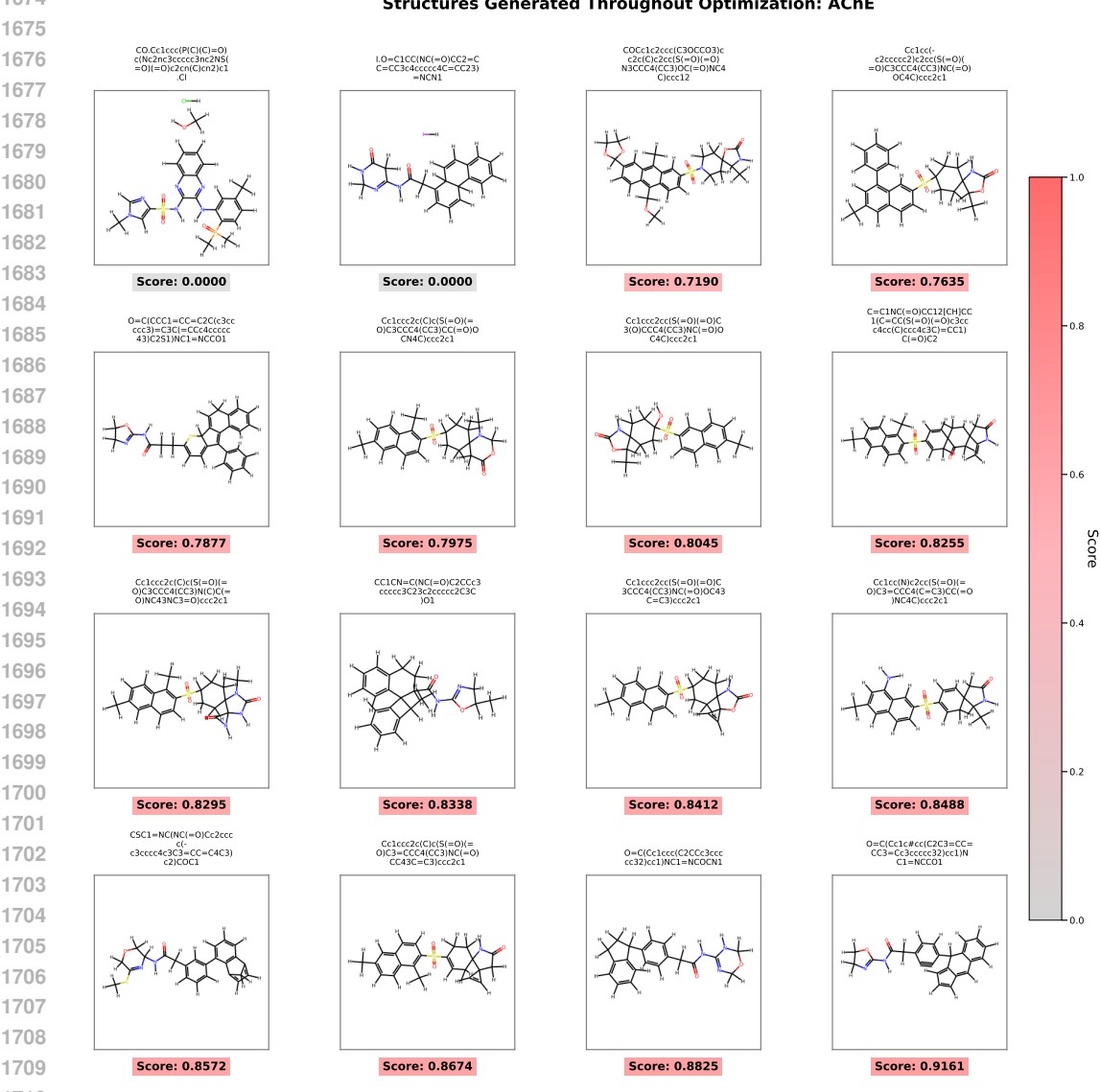

