# OpenReview forum: "Small Molecule Optimization with Large Language Models"
_ICLR.cc/2025/Conference — Submitted to ICLR 2025_

### Official Review · Reviewer_Hdqg · 2024-10-29

**Soundness:** 2
**Presentation:** 3
**Contribution:** 2
**Rating:** 6
**Confidence:** 4

**Summary:**

This paper presents a unique framework for optimizing small molecules using large language models (LLMs), specifically Chemlactica-125M, Chemlactica-1.3B, and Chemma-2B. These models, trained on over 100 million molecules extracted from PubChem, offer both generative and predictive capabilities tailored to molecular property prediction tasks. The primary contributions are as follows:
- Development of a Specialized Molecular Dataset: The authors construct a dataset rich in molecular properties, incorporating known structures, experimental properties, and optimized SMILES-based representations.
- Novel Optimization Algorithm: The paper introduces an innovative molecular optimization framework combining LLM-based generation with evolutionary strategies, specifically integrating genetic algorithms and prompt optimization techniques to enhance the molecular design pipeline.
- Benchmark Performance and Evaluation: By assessing performance on tasks such as Practical Molecular Optimization (PMO) and docking-based multi-property objectives, the authors claim state-of-the-art (SOTA) results across several key metrics, such as sample efficiency and the generative yield of viable molecules.

**Strengths:**

- Comprehensive Experimental Validation: The authors present a well-rounded suite of experiments, rigorously evaluating their models on various molecular design benchmarks, such as PMO, and providing comprehensive comparisons to current SOTA techniques across a diverse range of metrics.
- Innovative Optimization Framework: The integration of evolutionary strategies within a language-model-driven molecular generation pipeline is novel, blending genetic search concepts with LLM-specific prompt engineering. This hybridization is particularly suited to address the combinatorial complexity of chemical space.

**Weaknesses:**

- Inclusion of Baseline Comparison with Similar Methods:
The paper would benefit from a direct comparison with Wang et al.’s 2024 study, "Efficient Evolutionary Search over Chemical Space with Large Language Models," as this work also applies genetic algorithms with LLMs, albeit without fine-tuning. Since the authors acknowledge Wang et al.’s work as the most similar, a side-by-side comparison would strengthen the argument for the advantages of the current approach and illustrate any tangible benefits from fine-tuning.

- Model Selection and Justification:
While Chemlactica and Chemma are based on Galactica and Gemma models, it remains unclear why these were chosen over other more established and widely benchmarked models like Llama or GPT. Galactica and Gemma are comparatively limited in LLM applications, so examining the performance difference with a model like Llama or GPT, which are better validated, would be beneficial. This could help address concerns about model architecture suitability and provide insights into optimizing architectures for molecular tasks.

- Risk of Data Leakage and Benchmark Validity:
A significant limitation arises from the potential data leakage inherent in using PubChem-derived training data. Given that PMO and docking benchmarks aim to rediscover known drugs as a proxy for drug discovery, these molecules may already exist within PubChem. This overlap risks inflating performance metrics by providing the model with information it may have encountered during training. A clearer methodology or additional validation on a benchmark explicitly excluding known molecules from PubChem could address this confounder and validate the robustness of the framework.

- Task Selection and Generalizability:
The paper reports results on a subset of benchmark tasks (5 of 23 in PMO and 3 of 17 in MoleculeNet). This limited selection raises questions about the generalizability of the results, especially given that the ADMET prediction results in the appendix were less successful. Expanding the evaluation across additional tasks, or clarifying the criteria for task selection, could help establish confidence in the model’s consistency and overall applicability to various molecular optimization challenges.

### Reference
[1] Wang, Haorui, et al. "Efficient evolutionary search over chemical space with large language models." arXiv preprint arXiv:2406.16976 (2024).

**Questions:**

How does the method perform on tasks beyond what is shown in the main text?

---

> ### Author Response · Authors · 2024-11-22
>
> Thank you for the review, which raised concerns about baselines, modeling decisions, data quality, and the proposed method’s robustness. We address the points in turn below.
>
> **Inclusion of Baseline Comparison with Similar Methods**
>
> The reason we omitted a comparison with "Efficient Evolutionary Search over Chemical Space with Large Language Models” in our work is that the paper did not provide an evaluation of all of the tasks from the practical molecular optimization benchmark, making a direct and fair comparison with the remaining methods difficult. We provide below a direct comparison for the tasks that they provided in their analysis:
>
> [Link to the table](https://ibb.co/TM20Hbt)
>
> Our largest model Chemma-2B is on par with the MolLEO based on the closed GPT-4 model (the difference in the total score is less than the standard deviation), while even our smallest model Chemlactica-125M is significantly stronger than MolLEO based on the open models. With a closer look we see that our approach is dominating in multi-property optimization tasks, while MolLEO on GPT-4 is better at similarity/rediscovery and SMARTS based tasks.
>
> **Model Selection and Justification**
>
> We acknowledge that the robustness of the method’s performance to the underlying model was not extensively evaluated, primarily due to the computational cost of optimizing, tuning, and executing the continued pretraining protocol for the models, which were trained on dozens of billions of tokens. This method shows strong performance in both Gemma and Galactica architectures.
> We selected the models Galactica and Gemma as a basis of our work based on a combination of domain-specific knowledge, generally strong performance on standard benchmarks, as well as availability of variants with under 7B parameters. We describe this reasoning at the start of Section 4. The pretraining of the Galactica models included a lot of domain-specific data of interest in that it had been trained on a scientific corpus and had already seen SMILES representations enclosed in special tags, which we leverage in our optimization algorithm. Additionally, the Galactica models were implemented using the OPT architecture, which is virtually identical to GPT3.
> Thus, the Galactica models primarily differ from GPT in training data composition and pretraining methodologies, which abates concerns regarding the generalizability of our findings with respect to model architecture suitability.
> The Gemma models were not pretrained on a corpus of particular relevance to our work but demonstrated the strongest results on general-purpose language modeling benchmarks for their size. Given that LLama3 was not available at the time of our pretraining, and the Llama2 variant with the fewest parameters was LLama-2-7b, the Gemma-2b model was the strongest model available within our compute budget, (as we train on ~40B tokens). If there are other questions or points of clarification, we would be happy to elaborate on our reasoning.
>
> **Risk of Data Leakage and Benchmark Validity**
>
> The model can generate molecules that it has already been exposed to in its training data during optimization, and we recognize this as a limitation of our evaluation. However, despite the fact that PubChem is biased towards molecules of general pharmaceutical interest, the model did not, in its pretraining data, observe evaluations of the oracle scoring functions that are used throughout the molecular optimization tasks. This means that this mapping is learned during optimization.
>
> In Figure 2, we demonstrate that our method generates molecules with increasing distance from our training dataset (PubChem) as the molecular optimization process progresses, demonstrating that in order to reach a high reward, the model must generate molecules that do not exist within the pretraining data.
> Current benchmarks do not explicitly measure the novelty of generations with respect to the training data. While we believe it would be a nice addition (along with many other improvements of the existing benchmarks), we leave the design of more sophisticated benchmarks to future work.
>
> **Task Selection and Generalizability**
>
> To clarify, we perform evaluation against all 23 tasks within the PMO benchmark, with the sum total comparison with prior SOTA methods presented in the rightmost column of Table 3. Table 10 in Appendix Section A.9 presents the results of all three models across all 23 tasks in detail. We additionally demonstrate results on two suites of docking multi-property molecular optimization tasks.
>
> We made an effort to clarify our results, provide additional comparisons and underline our motivations. If this response has been to your standard, please consider revising your review score. We are also open to additional discussion if there are outstanding questions, points of unclarity or criticism. Thanks again for the review.

---

> > ### Comment · Reviewer_Hdqg · 2024-11-25
> >
> > Thank you for addressing my questions and providing additional results. The comparison with MolLEO appears reasonable to me. While the proposed method did not achieve outstanding performance, the results are generally comparable to the best outcomes from MolLEO, which highlights its value.
> >
> > I understand the reasoning behind the choice of model, given the high costs associated with training large language models (LLMs). However, I still believe that the risk of data leakage is a valid concern for methods relying on pretrained LLMs. That said, I agree that developing a more rigorous benchmark can be left for future work.
> >
> > Thank you also for directing me to the comprehensive PMO results. The results seem satisfactory to me.
> >
> > Based on the response and additional results, I have raised my rate.

---

> > > ### Author Response · Authors · 2024-12-02
> > >
> > > Thank you for your thorough review and helpful comments on our manuscript, we appreciate that you have no outstanding concerns and have carefully addressed each of your previous suggestions. While we've implemented these revisions, we remain committed to enhancing the manuscript's quality. Could you provide more specific guidance on what additional modifications, clarifications or elaborations might elevate the manuscript's scholarly contribution or methodological rigor? We remain open to your insights regarding the remaining gaps in the work and are grateful for your commitment to the improvement of our work.

---

### Official Review · Reviewer_W49H · 2024-10-29

**Soundness:** 2
**Presentation:** 3
**Contribution:** 3
**Rating:** 6
**Confidence:** 4

**Summary:**

The paper introduces 3 models Chemlactica-125M, Chemlactica-1.3B, and Chemma-2B which fine-tunes Galactica. The new series of models are trained on a custom prepared dataset based on PubChem data. The final models were used for property prediction and molecular optimisation tasks. The models show strong empirical performance, outperforming or matching recent existing methods.

**Strengths:**

* The authors release the training corpus and model checkpoints
* Table 1 shows the benefit of transfer learning
* Property prediction experiments show strong performance
* Molecular optimisation experiments are thorough and compared to strong baselines
* The Appendix is detailed and the transparency around hyperparameter tuning, information around floating point precision is interesting

**Weaknesses:**

Generally, descriptions of the model and pre-training are thorough but there are important metrics and discrepancies in the pre-training dataset that should at least be discussed. I will combine the specific points and related questions in the Questions section.

**Questions:**

1. From the main text results, Chemlactica-125M, Chemlactica-1.3B, and Chemma-2B show strong empirical performance, matching or outperforming recent strong baselines. It is clear that the models work, but it is unclear how much of the benefit is from the pre-training data itself and leveraging the base pre-trained Galactica. I will focus my discussion here on the molecule optimisation experiments. On the PMO benchmark, the authors outperform all compared models by a wide margin. However, the PMO benchmark was designed with ZINC 250k as the pre-training data. All models in the benchmark were pre-trained with this data. How would the performance differ if all the existing models in PMO were pre-trained with PubChem and/or Chemlactica/Chemma were further pre-trained on ZINC 250k and not PubChem? Based on this, I had also checked the Augmented Memory (ChEMBL) [1], GEAM (ZINC 250k) [2], and Saturn (ZINC 250k) [3] papers cited by the authors and their pre-training data. There is existing literature suggesting that changing from ChEMBL to PubChem pre-training data alone improves performance considerably [4]. Pre-training data is expected to have a notable impact on optimisation performance since the models fit this data. While I can appreciate that part of the point of the models released by the authors is leveraging "big data", an ablation on fine-tuning Galactica with just ZINC 250k and/or ChEMBL and/or pre-training the comparison models with PubChem would enable a more thorough understanding of which component of the proposed models leads to the biggest performance improvement.

2. Table 9 shows high variability and seemingly unpredictable behavior when tuning for conditional generation. This section was shown for molecular weight but is this behavior also observed for other tasks? The authors state that the optimal hyperparameters are the same across the models but is this consistent with other tasks? Molecular weight is inexpensive to compute and if this tuning behavior is very sporadic, it might be difficult to control the behavior with other more expensive properties. It would be informative to show the tuning statistics on other properties.

3. Figure 11 shows the docking scores of molecules for the DRD2 experiment. It looks like there is high variance even towards the end of the run.  An advantage of generative models that are fine-tuned is focused modeling of a good distribution. I would have expected that the variance decreases as good molecules are found since the model is being fine-tuned with only the best molecules. Is this variability present on the other docking targets? I can appreciate that the models generate molecules with good docking scores and that this does not take away from that fact, but I am interested in hearing the author's thoughts on potential reasons for the variability. A potentially interesting experiment would be to purposely fine-tune the models with sets of very similar molecules. Does this still lead to high variability?

4. In Table 5/6, GEAM reports diversity with #Circles [5]. Do the authors also have these metrics?

5. Are there statistics on how many times fine-tuning was performed and how long this takes?

6. How much memory and time does it take to deploy and inference the models?

7. Minor comment: typo in number of valid candidate molecules 10e60?

Overall, the transparency in the paper and the strong empirical performance are positive points. The main questions I have are centered around how much benefit is from the pre-training data compared to the specific workflow introduced (Galactica fine-tuning on PubChem-derived dataset). I am happy to engage in discussions with the authors.


[1] Augmented Memory: https://pubs.acs.org/doi/10.1021/jacsau.4c00066

[2] GEAM: https://arxiv.org/abs/2310.00841

[3] Saturn: https://arxiv.org/abs/2405.17066

[4] REINVENT with Transformer: https://jcheminf.biomedcentral.com/articles/10.1186/s13321-024-00887-0

[5] #Circles: https://openreview.net/forum?id=Yo06F8kfMa1

---

> ### Author Response · Authors · 2024-11-23
> **Answers to Questions 1-2**
>
> We greatly appreciate the thorough review. We provide our responses to your numbered inquiries in the corresponding order.
>
> 1. .
>     1. We acknowledge the challenge of disentangling the benefits of large-scale data, model architectures, and optimization methodology. The datasets involved — ZINC 250k, ChEMBL, and PubChem—differ significantly in size, diversity, and representational richness. For instance, prior analysis has shown that PubChem contains 1,281,788 unique Murcko ring with linker scaffolds, which exceeds the number of molecules in Zinc250K by an order of five times [1]. Supported by the observed positive relationship between training data scale and conditional generation performance in Table 1. (Chemma-2B trained on 2.1 billion vs 39 billion tokens), we hypothesize that this increased diversity contributes significantly to our results, although this effect is inextricably connected with our approach’s capacity to leverage this scale of data effectively.
>     2. There are two ways we could entangle the data and the model architecture. We could try to (i) train Galactica and Gemma on a corpus produced from ZINC-250K, and (ii) replace ZINC-250K with PubChem in the baselines. For (i) we do not expect the models to learn any meaningful notion of similarity with ~500x less tokens, which is critical for the genetic algorithm. For (ii) we expect most of the small scale models will saturate quickly and will not be able to leverage 110M+ molecules. (e.g. Augmented Memory uses a 3-layer LSTM with less than 2M parameters). This implies that both “dimensions” play a critical role. A similar expectation comes from the intuition from the scaling laws that suggests proportional scaling of data and network size is needed to maintain optimality. This implies that scaling only one of the two (i.e. data or network size) by 2.5 orders of magnitude would lead to suboptimal results. Proper experimental verification of these expectations would require extensive tuning of hyperparameters, which we thought is beyond the scope of this work.
>     3. To more directly address questions related to the optimization method, the comparison with the MolLEO baseline leveraging GPT-4 shown below demonstrates the strength of our approach in balancing model, data, and algorithmic scalability. Despite GPT-4 having a larger pre-training corpus and parameter count than our models, our method leveraging the 1.3B parameter Chemlactica model outperforms it on most tasks they report.     [Link to the table comparing MolLEO and our models](https://ibb.co/TM20Hbt)
>     4. In conclusion, thoroughly evaluating molecular optimization methods requires scaling pre-training datasets and adapting models to handle these scales. To this end, we encourage future work to expand on our findings and provide all pre-trained models and datasets for public use. We hope our work inspires further exploration of how different datasets, model architectures, and optimization algorithms interact.
>
>
> 2. We executed an equivalent hyperparameter search for conditional generation using the synthetic accessibility score (SAS) to compare to the results we presented on weight in the main text. The best-performing hyperparameters were the same as for weight. These results suggest that this hyperparameter configuration is at least near-optimal for conditional generation on simple molecular properties. Note that “DNF” is an acronym for did not finish, as the model either repeatedly generated the same substring or continued to generate invalid smiles such that the trial did not finish in a reasonable time. The remaining runs spanned three to four minutes, and the DNF trial exceeded 30 minutes of runtime when it was manually terminated.
>
>     [Link to the hyperparameter search table for SAS-conditioned generations](https://ibb.co/sjksRMZ)
>
> [1] Velkoborsky, J., & Hoksza, D. (2016). Scaffold analysis of PubChem database as background for hierarchical scaffold-based visualization. Journal of Cheminformatics, 8, 1-14.

---

> > ### Author Response · Authors · 2024-11-23
> > **Answers to Questions 3-7**
> >
> > 3. A couple of potential explanations exist for increased variability in generations near the end of optimization. The most likely contributing factor is the dynamic temperature scheduling throughout optimization. This causes the model to generate more diverse molecules towards the end of the optimization process. As the temperature is highest at the very end, it is expected that SMILES with low likelihoods will be generated, with both very high VINA and low VINA scores. Another less likely explanation is that as the model reaches the Pareto frontier for the multi-property optimization task, the model generations increasingly exploit the non-docking properties to increase the reward because they were easier to learn throughout the finetuning rounds. Finally, it is important to remember that molecules that did not dock were not included in Figure 11. The number of non-docking molecules increases for higher temperature generations, and this relationship may create an impression of increased spread at the end of optimization due to greater sparsity.
> > 4. We calculated the *#Circles* metric for molecules satisfying the Strict Hit Ratio conditions upon request. The baselines did not report this metric for molecules that satisfy the more relaxed Hit Ratio criteria, so we do not present those results here. We display the results for our methods alongside the baselines to facilitate comparison. The results demonstrate that our approach generates more diverse high-reward molecules than other methods.
> >
> >     [Link to the table with the comparisons according to the #Circles metric](https://ibb.co/cJt3yQQ)
> >
> > 5. A couple of factors influence the span of training during optimization, namely the size of the data and the number of epochs. Since we use all of the molecules in the Pool, which is around 30, and train for five epochs, training is relatively short. With the hyperparameters we used, the average training time for a single round of finetuning during multi-property optimization in docking tasks takes approximately one to two minutes on a single 40Gb A100 GPU. We do not exceed ten rounds of finetuning in a single optimization process.
> > 6. The amount of time required for the deployment of the model depends on several factors. First, what is the task that the models are being applied to and what are the computational resources available. With respect to hardware used, the number of available CPU cores for docking simulations and other score calculations affect the wall time. Similarly, the the number and model of GPUs available to run inference on. We measured the time of **a single docking experiment (with one seed). Using Chemlactica-125M on a single 40Gb A100 GPU takes approximately 3 hours. Notably, 80% of this time was consumed by the docking simulation, which is independent of our proposed method, meaning model inference and training alone required ~40 minutes.** The amount of memory used during inference depends on the model used, the batch size of generations and the precision of the model. For our experiments, we tuned the generation batch size such that we used the 40Gb of GPU memory available in our systems. We provide a table below which presents the GPU usage for the chemlactica-125m model loading and memory used for inference by taking the difference between peak CUDA memory allocation at inference time and model loading alone. We used 2048 tokens as input because this is the effective context length of our models. Memory used for batched inference scales linearly with the number of inputs passed to the model in parallel.
> >
> >
> >     | Model | Loading Model (FP32) | Loading Model (BF16) | 2048 token inference (FP32) | 2048 token inference (BF16) |
> >     | --- | --- | --- | --- | --- |
> >     | Chemlactica-125M | 477.66 MB | 245.48 MB | 582.63 MB | 295.13 MB |
> >     | Chemlactica-1.3B | 5018.47 MB | 2509.23 MB | 1888.00 MB | 944.00 MB |
> >     | Chemma-2B | 9560.30 MB | 4780.15 MB | 6079.24 MB | 3044.00 MB |
> > 7. Thank you for catching this, that’s a typo which should be 10e60. We’ll make sure to amend this in the revised version.
> >
> > Your detailed feedback was invaluable, and we hope our responses align with your expectations. Should there be any outstanding uncertainty or reservations, we would be happy to discuss any issues or answer new questions. We would appreciate an updated assessment if you believe that the above responses have strengthened the motivation for this work. We will update the manuscript to incorporate parts of these answers.

---

> > > ### Comment · Reviewer_W49H · 2024-11-25
> > > **Response to authors**
> > >
> > > Thank you for the detailed response and interesting insights and for running extra experiments. Overall, I want to acknowledge the transparency in all aspects of the model, from pre-training to hyperparameter tuning, to deployment. The authors have answered most of my questions (and also thank you for the additional results) and I will follow-up only on the point which I still am uncertain about.
> > >
> > > ### PubChem pre-training
> > >
> > > Overall, I agree with the authors that using a more diverse dataset is important and that their method can leverage this scale. Coupled with the strong empirical performance, I do believe the proposed Chemlactica/Chemma models have benefits beyond just using a larger dataset than ZINC 250k and ChEMBL. Regarding the authors' statement that scaling both the architecture and data **together** is important, I agree. However, I want to note that one of the papers that has shown moving to PubChem yields benefits [1], should be < 25M parameters which is considerably smaller than the billion-parameter scale models. While I do not know if this model's size *leverages* the PubChem dataset optimally, it shows that a relatively small model immediately has large benefits moving beyond ZINC and ChEMBL. Relatedly, I still believe the use of different pre-training data in the experiments is a big caveat. From the previous response, PMO was proposed with the ZINC 250k pre-training data. The compared MolLEO method also initialises their GA population with ZINC 250k sampled molecules. I acknowledge the authors response about GPT4 in that work being *much* larger than Chemlactica/Chemma but using it as inference through the API imposed no pre-training or fine-tuning costs.
> > >
> > > However, I do believe the insights from the paper are valuable and again, the transparency is really appreciated. **I raised the score to 6 to be above acceptance.**
> > >
> > > [1] https://jcheminf.biomedcentral.com/articles/10.1186/s13321-024-00887-0

---

> > > > ### Author Response · Authors · 2024-12-02
> > > >
> > > > We are gratified by your recognition of the merits of our work and your engagement with our response. We will now turn to the outstanding concern regarding pretraining data consistency. We’d like to emphasize that although the additional cost of pretraining and finetuning in our method when compared to MolLEO+GPT-4 baseline is a valid observation, this argument is entirely orthogonal to the discussion about dataset size/diversity and model size. Regardless of how the base model is applied, the MolLEO+GPT-4 baseline had a larger dataset and used a larger model. Thus, our stronger performance on this benchmark is meaningful in disentangling the performance of our method from the size of the models and dataset used. Additionally, note that the MolLEO work did not run their method on all tasks in the practical molecular optimization benchmark. We believe this provides evidence against the remaining concern with the work.
> > > >
> > > > Additionally, we can perform an experiment training a model on Zinc-250k and compare with the baseline methods we include for PMO to provide further clarity on this matter. If this experiment would significantly strengthen our work, we would be happy to furnish the results and discuss further.

---

### Official Review · Reviewer_SPuw · 2024-11-02

**Soundness:** 4
**Presentation:** 3
**Contribution:** 3
**Rating:** 6
**Confidence:** 4

**Summary:**

This paper explores small molecule optimization using Large Language Models (LLMs) based on evolutionary strategies. The authors introduce pair tokens to represent molecules and utilize these tokens to express various properties and SMILES strings. They fine-tune a pre-trained model using this token-based approach and perform molecule optimization and property prediction. The study demonstrates state-of-the-art performance in molecule property prediction on specific datasets through supervised fine-tuning. Additionally, the authors propose a novel SMILES generation method inspired by genetic algorithms using their token system. Finally, they present a molecule optimization algorithm based on dynamic fine-tuning, which achieves state-of-the-art results on molecule optimization benchmarks.

**Strengths:**

1. The study successfully demonstrates the feasibility of molecule optimization using LLMs and a special token system.
2. The innovative approach of emulating genetic algorithms through the token system and Chain of Thought reasoning is particularly noteworthy.

**Weaknesses:**

### Lack of Computational Efficiency Comparison
1. The paper does not provide a comparison of overall processing times between methods.

### Exploration of Efficient Training Methods
1. Have the authors considered more efficient learning methods beyond fine-tuning the entire model?
2. It would be interesting to explore the effects of techniques such as freezing specific layers, layer skipping, or parameter-efficient fine-tuning.

### Limited Exploration of LLM Capabilities for Multi-Property Optimization (MPO)
1. Additional experiments demonstrating the potential of LLMs for MPO would be beneficial.

**Questions:**

### Lack of Computational Efficiency Comparison
1. Given that LLM inference can be computationally expensive, it would be valuable to know the time required for molecule optimization. Is it possible to compare this with other models?

### Exploration of Efficient Training Methods
1. Can you explore the effects of techniques such as freezing specific layers, layer skipping, or parameter-efficient fine-tuning?

### Limited Exploration of LLM Capabilities for Multi-Property Optimization (MPO)
1. How would the performance change if target properties were enumerated rather than produced as a product?

These points could be addressed to further strengthen the paper and provide a more comprehensive understanding of the proposed method's capabilities and limitations.

---

> ### Author Response · Authors · 2024-11-23
>
> Thank you for a considerate and thorough review, we address the limitations and questions asked in separate parts below.
>
> **Lack of Computational Efficiency Comparison**
>
> Despite the potentially high computational cost of LLM inference and the compute requirements of other molecular optimization methods, applications of computation methods in drug discovery are not typically constrained by computational cost. This is because the computation required to generate candidates is assumed to be less costly than the oracle, which would be a wet lab test in industrial use cases. Because oracles are prioritized over computational cost, the benchmarks included in the present work focus on sample efficiency rather than FLOP optimality or compute wall time.
>
> To provide some concrete numbers, **a single docking experiment (with one seed) using Chemlactica-125M on a single 40Gb A100 GPU takes approximately three hours. Notably, the docking simulation consumed 80% of this time, which is independent of our proposed method (assuming synchronous flow), meaning model inference and training alone required ~40 minutes.** We do not provide a comparison with other methods as they did not make this information publicly available, and a comprehensive evaluation of the compute costs of molecular optimization methods (via our re-implementation) is beyond the scope of our work. We will add these numbers to the manuscript.
>
> Finally, as was correctly identified in the concern addressed by the subsequent section, an advantage of our method is that it leverages large language models and thus inherits a broad range of techniques for more efficient training and inference, including different inference engines, parameter finetuning methods and quantization, among others. Thus, should computational cost become a bottleneck, our approach has an array of promising directions for further exploration to ameliorate such issues.
>
> **Exploration of Efficient Training Methods**
>
> As motivated in the previous section, this kind of analysis is generally beyond the bounds of this work, in which we primarily seek to demonstrate the effectiveness of a novel molecular optimization algorithm developed on continually pre-trained LLMs. Even so, we analyze the effects of running optimization with mixed precision, as it is one of the most ubiquitous methods used for more efficient training and inference. We show that lower precision training and inference can sometimes impede performance. Given our specific use case in which we have sufficient tokens and are not concerned with computational cost, it is preferable to do a full training of the entire model instead of using optimized methods, as they almost always incur a cost in task-specific metrics.
>
> To explore this point further, we extended the experiment on the rat liver microsome (RLM) SFT, where we compared the performance and training time of the model with full finetuning and freezing 75% of the layers. The run time was decreased by 8.3% of the full finetuning. However, the model's performance was significantly worse, demonstrating poorer performance than the adme-fang-RandomForestRegressor, which all of our fully finetuned models outperformed. There appears to be a clear tradeoff between efficiency and performance in molecular-specific adaptation during finetuning. This result further supports the abovementioned analyses in the text on training and inference in half-precision.
>
> |  | RLM - pearson R | run time (20 epochs) |
> | --- | --- | --- |
> | full pretraining (HP tuned) | 0.72 | 3:36 mins |
> | 75% frozen (HP from best full tuned) | 0.61 | 3:18 min |
> | 75% frozen (HP tuned) | 0.62 | 3:18 min |
>
> **Limited Exploration of LLM Capabilities for Multi-Property Optimization (MPO)**
>
> Our method should extend to the explicit optimization of multiple properties through their enumeration in prompts. We provide limited evidence in Appendix Section A.9, which shows that including individual learned properties in prompts during optimization improves performance on most task-model combinations we tried.
>
> In the current benchmarks for molecular optimization, the reward scores are implemented as scalarized combinations of individual scores. Thus, we do not alter the design of our method to account for potential access to multiple properties. Additionally, we do not look towards the challenges in the so called “multi-objective optimization” setting, such as determining Pareto optimality. We acknowledge that better benchmarks in future might consider these aspects as well.
>
> We thank you again for your review focused on the potential for computational efficiency and flexibility in the LLM paradigm. We trust that our responses have adequately addressed your concerns with reasoning, data and experiments, as necessary. We hope that our revisions merit a positive re-evaluation of our work.

---

> > ### Author Response · Authors · 2024-12-02
> >
> > We appreciate the opportunity to further clarify the scientific merits of our work. Given the thorough nature of our response, we respectfully request that you indicate if any of our explanations remain unsatisfactory, or reconsider the current rating in light of the clarifications provided. We are committed to scientific rigor and collaborative development of our work, thus we look forward to your continued engagement and evaluation.

---

### Official Review · Reviewer_Hdmr · 2024-11-03

**Soundness:** 2
**Presentation:** 3
**Contribution:** 2
**Rating:** 5
**Confidence:** 3

**Summary:**

This paper presents a novel approach to molecular optimization for drug discovery by leveraging the large language models. The contribution are included as follows, LLM-based molecular design, new molecular corpus, optimization algorithm and this paper also claims achieves the state of art performance.

**Strengths:**

The strengths of the paper lie in several key areas.
1. comprehensive dataset, this authors created a custom molecular corpus with over 100 million molecules from PubChem, incorporating detailed chemical properties.
2. The combination of LLMs with a genetic algorithm, prompt optimization, and rejection sampling allows the paper’s method to effectively explore chemical space and optimize for multiple properties at once.
3. Versatility: The models demonstrate adaptability, achieving high performance even with minimal fine-tuning data, which underscores their capability with limited datasets.
4. Open Access: The authors prioritize reproducibility by openly sharing their training data, models, and optimization algorithms with the research community.

**Weaknesses:**

The weakness of this paper includes:
1. The model only consider the smile representation, it lacks explicit consideration of 3D conformation
2. The proposed optimization algorithm, while efficient, still relies on a high number of oracle evaluations. This paper could further improve reduce oracle calls, especially for applications where computationally intensive evaluations may be costly.
3. Limited experimental validation: while the paper demonstrates strong results on computational benchmarks, it would benefit from additional validation in real-world experimental settings, such as testing generated molecules in biological assays.
4. The performance of the models appears sensitive to hyperparameter choices, especially during the optimization process.
5. The molecule optimization algorithm lacks of novelty.
6. Missing references to recent studies.

**Questions:**

see weakness

---

> ### Author Response · Authors · 2024-11-21
>
> Thanks for raising considerations regarding the scope of our work and concerns about method’s effectiveness and novelty. We’ll address your concerns point by point:
>
> 1. It is true that the model only considers the SMILES representation of molecules rather than any explicit modeling of the 3D conformation state of the molecule. There are two extenuating factors explaining this. Firstly, the primary goal of the method is to optimize for “better” molecules within chemical space rather than conformations. Secondly, the reason for this direction is that the currently available benchmarks for evaluating molecular optimization methods are all SMILES-based. To elaborate, none of the practical molecular optimization (PMO) benchmark tasks are conformation sensitive, and all docking simulations change the input conformation during pose estimation. This lack of consideration for conformations is a limitation that applies to the current state of this field, and we agree that better, 3D-aware benchmarks would help guide more practical research in this space. However, designing such benchmarks requires careful oracle design, which ensures accuracy, generalizability, and computational feasibility; thus, the creation of novel benchmarks for molecular optimization is beyond the scope of this work.
> 2. Further improvements in sample efficiency would make the proposed method more applicable to industrial use cases, where oracle scores are more computationally costly. We believe this work is a step in exactly that direction: it presents a novel method that achieves state-of-the-art results across several molecular optimization tasks. For certain benchmarks (e.g. docking on DRD2, ACHE, and MK2) we got 3-4x improvements of sample efficiency compared to the prior art.
> 3. In a drug-discovery campaign, an LLM in-the-loop method should be used with wet lab testing as the oracle. We recognize that lack of wet-lab validation is a limitation of the present work, but lack of infrastructure and funding prohibits it. In addition, this would preclude us from comparing with other methods, none of which performed this kind of evaluation and analysis.
> 4. We recognize concern about the sensitivity of models to hyperparameter selection and acknowledge it as a limitation of the work. This aspect of the method's performance is explicitly presented via multi-seed runs and visual representations of optimization trajectories. Additionally, note that hyperparameters selected based on the two PMO tasks (zaleplon_mpo and perindopril_mpo, consistent with the PMO paper) were then used for the remaining PMO tasks and the DRD2, ACHE, and MK2 docking MPO tasks. Similarly, the illustrative experiment from Saturn was used to select hyperparameters for the remaining tasks in the benchmark. These results demonstrate that although the proposed method's performance is sensitive to hyperparameters, hyperparameter selection transfers well across diverse property optimization tasks.
> 5. Despite the existence of various methods for molecular optimization using large language models, which we enumerate in our related work section, our model demonstrates novelty in that it is the only work to combine iterative LLM finetuning, genetic algorithms, and discrete prompt tuning. In addition, it is the first work (as far as we know) that combined iterative language model training with evolutionary methods. The algorithm's novelty stems from combining a suite of disparate methods into an end-to-end pipeline that demonstrates strong performance in a variety of scenarios.
> 6. We would be happy to amend our related work upon recommendation of any relevant literature that we have omitted. The paper currently includes an extensive enumeration of related literature, covering large language models for chemistry, adjacent methods in molecular optimization, and work at the intersection of these domains.
>
> We hope we addressed all of the concerns you raised to your satisfaction. If that is the case, we would ask to adjust the review score accordingly. We are open for more questions and feedback. Thanks again.

---

> ### Comment · Reviewer_Hdmr · 2024-11-27
>
> Thank you for your response, which addressed part of my concerns and led me to increase my score. However, I still have some concerns regarding novelty and references.
>
> While your work introduces a molecular optimization algorithm and claims to be the first to combine "iterative" language model training with evolutionary methods as the main contribution, I noticed that a recent study, LM-Design [1], also utilized LLM and iterative refinement for protein design. This overlap in approach raises questions about the novelty of your work. This LM-Design work is missing reference and discussion.
>
> Furthermore, in the Molecule Optimization Techniques, I found that other recent studies which seems quite relevant but not referenced or discussed, such as DrugImprover [2] and ReLeaSE [3], a RL-based molecule optimization algorithm, and DrugAssist [4], a Large Language Model for molecule optimization algorithm.
>
> These omissions limit the clarity of your work's contributions. If you address these above concerns and properly discuss the relevant studies in your updated version, I will consider to further increasing my score.
>
> [1] Structure-informed Language Models Are Protein Designers, ICML 2024
> [2] DrugImprover: Utilizing Reinforcement Learning for Multi-Objective Alignment in Drug Optimization, NeurIPS 2023 Drug Discovery Workshop
> [3] Deep reinforcement learning for de novo drug design, Science 2018
> [4] DrugAssist: A Large Language Model for Molecule Optimization, arXiv 2023

---

> > ### Author Response · Authors · 2024-12-02
> >
> > Thank you for your continued engagement with our work, and for providing an opportunity to clarify how our work contrasts from the existing literature in the field. We have revised our submission to include comparisons with the work you suggested. We include the relevant comparisons here for your reference.
> >
> > Structure-informed Language Models Are Protein Designers presented a method for protein design optimization, leveraging pre-trained protein language models without additional finetuning. Deep reinforcement learning for de novo drug design combined iterative training with small RNNs for molecular optimization; despite the title, evolutionary methods are not applied. DRUGIMPROVER: Utilizing Reinforcement Learning for Multi-Objective Alignment in Drug Optimization makes use of an RNN as a generator rather than an LLM, performs finetuning of the generator policy once, and does not leverage evolutionary methods. Finally, DRUGASSIST: A LARGE LANGUAGE MODEL FOR MOLECULE OPTIMIZATION, only performs finetuning once and does not leverage evolutionary methods.
> >
> > The distinguishing characteristics of our method include its ability to operate at a large scale by leveraging LLMs which we trained on a specialized corpus, the use of an evolutionary method that allows us to transfer understanding of molecule similarity for application in arbitrary black box optimization tasks and iterative finetuning throughout optimization that allows the model to learn an increasingly narrow and specific chemical subspace of interest.

---

### Meta-Review · Area_Chair_cHtv · 2024-12-19

**Metareview:**

The paper studies the usage of Large Language Models (LLMs) for the generation and design of molecular systems. Namely, the paper contributes to the design in three ways: it curates a part of PubChem for further usage as a finetuning dataset; it finetunes Galactica and Gemma for property prediction and conditional generation; it leverages genetic algorithms for property optimization. One of the main contributions of the current work is that all three contributions are made publicly available, which was highlighted by the reviewers.

Overall, the paper is rated positively by reviewers but without anyone championing its acceptance. The common concern among the reviewers can be summarized as the lack of a clear takeaway message. Indeed, data curation, finetuning, and optimization during inference have been previously studied in the same or similar contexts and are accepted as techniques for performance improvement. Thus, it is not clear what the main message of the paper would be for the ICLR audience.

**Additional Comments On Reviewer Discussion:**

The reviewers mostly raised concerns regarding evaluations and experimental setups. I would like to highlight the following points of the discussion that would potentially strengthen the paper.
1. Ablation study of the role of finetuning (proposed by Reviewer W49H). This would allow the authors to motivate their design choices and potentially gain new insights into how to further improve the proposed approach.
2. Ablation studies of different models (proposed by Reviewer Hdqg). Despite being computationally expensive, meticulous experimental evaluations are necessary for empirical studies.
3. Addressing the issues with potential data leakages for evaluation (proposed by Reviewer Hdqg). This direction would allow the authors to reconsider their main goal, i.e. if it is data retrieval, then data leakage is irrelevant. On the other hand, if it is a discovery of new molecules, then great care has to be taken when fine-tuning models.

---

### Decision · Program_Chairs · 2025-01-22

Reject